# Sequences and proteins that influence mRNA processing in *Trypanosoma brucei*: Evolutionary conservation of SR-domain and PTB protein functions

**Albina Waithaka[1], Olena Maiakovska[2], Dirk Grimm[2,3], Larissa Melo do Nascimento[1], Christine Clayton[1]***

**1** Heidelberg University Centre for Molecular Biology (ZMBH), Heidelberg, Germany, **2** Department of Infectious Diseases/Virology, Section Viral Vector Technologies, Medical Faculty, University of Heidelberg, BioQuant, Heidelberg, Germany, **3** German Center for Infection Research (DZIF) and German Center for Cardiovascular Research (DZHK), partner site Heidelberg, Heidelberg, Germany

* cclayton@zmbh.uni-heidelberg.de

## Abstract

### Background

Spliced leader *trans* splicing is the addition of a short, capped sequence to the 5' end of mRNAs. It is widespread in eukaryotic evolution, but factors that influence *trans* splicing acceptor site choice have been little investigated. In Kinetoplastids, all protein-coding mRNAs are 5' *trans* spliced. A polypyrimidine tract is usually found upstream of the AG splice acceptor, but there is no branch point consensus; moreover, splicing dictates polyadenylation of the preceding mRNA, which is a validated drug target.

### Methodology and principal findings

We here describe a *trans* splicing reporter system that can be used for studies and screens concerning the roles of sequences and proteins in processing site choice and efficiency. Splicing was poor with poly(U) tracts less than 9 nt long, and was influenced by an intergenic region secondary structure. A screen for signals resulted in selection of sequences that were on average 45% U and 35% C. Tethering of either the splicing factor SF1, or the cleavage and polyadenylation factor CPSF3 within the intron stimulated processing in the correct positions, while tethering of two possible homologues of Opisthokont PTB inhibited processing. In contrast, tethering of SR-domain proteins RBSR1, RBSR2, or TSR1 or its interaction partner TSR1IP, promoted use of alternative signals upstream of the tethering sites. RBSR1 interacts predominantly with proteins implicated in splicing, whereas the interactome of RBSR2 is more diverse.

### Conclusions

Our selectable constructs are suitable for screens of both sequences, and proteins that affect mRNA processing in *T. brucei*. Our results suggest that the functions of PTB and SR-

**Data Availability Statement:** DNA sequencing data are available at Array express with accession numbers E-MTAB-11647 (preliminary PPT screen);

E-MTAB-11628 (2nd PPT screen); E-MTAB-11627 (RBSR1 RNAi); and E-MTAB-11648 (RNSR2 RNAi). The mass spectrometry proteomics data have been deposited to the ProteomeXchange Consortium via the PRIDE partner repository with the dataset identifier PXD033509. Excel tables for the graphs are at 10.6084/m9.figshare.21100483. Full gel images are in the main paper or the supplementary material.

**Funding:** This work was partially funded by Deutsche Forschungsgemeinschaft grant number CI112/26-1 to CC, and by core support to CC from the state of Baden-Württemberg. Both sources partially paid salaries to AW and LN. The funders had no role in study design, data collection and analysis, decision to publish, or preparation of the manuscript.

**Competing interests:** The authors have declared that no competing interests exist.

domain proteins in splice site definition may already have been present in the last eukaryotic common ancestor.

## Author summary

Spliced leader *trans* splicing is the addition of a short sequence to the 5' end of mRNAs. It is widespread in eukaryotic evolution, but factors that influence the splicing position have been little investigated. In trypanosomes, all protein-coding mRNAs are 5' *trans* spliced. Moreover, splicing dictates polyadenylation of the preceding mRNA, which is a validated drug target. We here describe a new reporter system, which we used for a screen to define sequences needed for *trans* splicing. We also artificially bound a variety of proteins just upstream of the splicing signal and found that some stimulated splicing, some inhibited it, and some moved mRNA processing to a new position.

## Introduction

Processing of nuclear-encoded mRNAs by 5' *trans* splicing of short capped "spliced leader" sequences is scattered throughout eukaryotic evolution, being found in at least three different supergroups [1]. The mechanism is overall similar to that of *cis* splicing; and while it is usually assumed to have evolved independently numerous times, there is also a possibility that it was present in the common ancestor and subsequently lost [1]. In comparison with *cis* splicing, both the sequences required for *trans* splicing, and the factors that regulate it have been relatively little investigated.

In *Trypanosoma brucei* and related kinetoplastid parasites, dependence on *trans* splicing is extreme, since transcription of nearly all protein-coding genes is polycistronic. Trypanosomes therefore serve as a useful model system to study *trans* splicing. The mature capped 5'-ends are formed by addition of the capped spliced leader (*SL*) (reviewed in [2,3]). The spliced leader precursor mRNAs, which are called *SLRNA*s, include the *SL* sequence and a short intron. The *SLRNA* assembles into a ribonucleoprotein particle (RNP) [4] which is recruited with other spliceosomal components to the splice acceptor sites of pre-mRNAs (Fig 1A). As in other eukaryotes, splicing factors are thought to recognize a polypyrimidine tract (PPT) upstream of the splice site. The spliced leader intron forms a Y structure by 2'-5' linkage with any A residue immediately upstream of the PPT; there is no branch-point consensus sequence [5]. Polyadenylation occurs approximately 100 nt upstream of the PPT. There is no polyadenylation signal within the 3'-untranslated region (UTR) but A residues are preferred at the cleavage site [6–9]. Subsequently the Y is debranched and the intergenic sequence is discarded. Splicing and polyadenylation are physically and mechanistically linked: if splicing is prevented, polyadenylation also cannot occur, and *vice versa* [10–14]. The components of the splicing and polyadenylation complexes have been characterized [15–22] but the nature of the connection between them is still not understood. Notably, one component of the polyadenylation machinery, the cleavage and polyadenylation factor CPSF3 (also called CPSF73), is the validated target of candidate benzoxaborole drugs for treatment of both human and animal trypanosomiases [13,14].

Two life-cycle stages of *T. brucei* grow well *in vitro*: the mammalian "bloodstream" form, and the "procyclic" form which grows in the midgut of Tsetse flies. The sequence requirements for kinetoplastid *trans* splicing have been investigated in various ways. Many of the initial experiments were done by transient transfection of reporters into procyclic forms, followed by

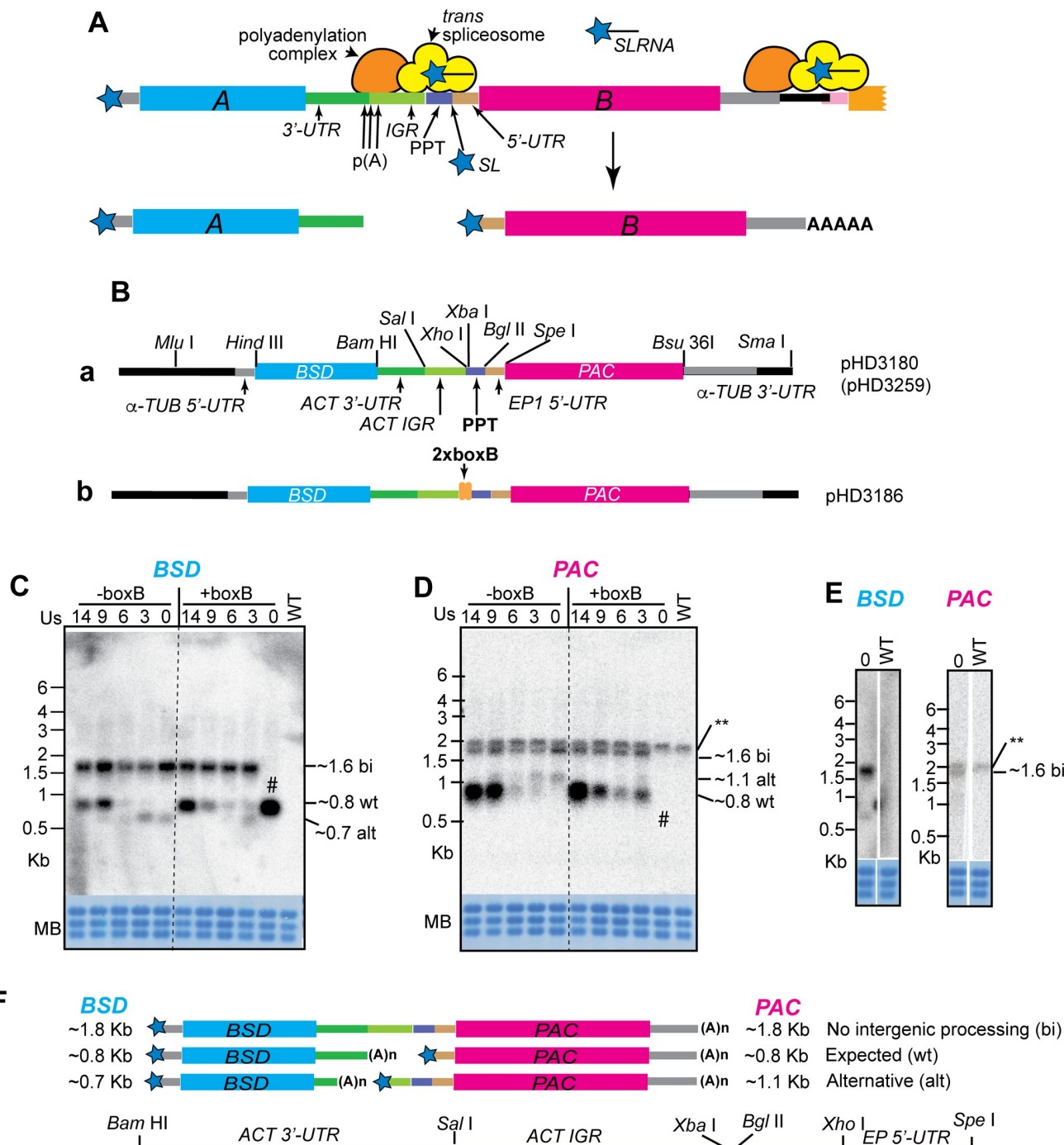

**Fig 1. Effect of PPT length and a secondary structure on mRNA processing.** A. Scheme showing the mechanism of *trans* splicing and polyadenylation. Important players and sequence regions are labelled. B. Maps of the two vectors used including important restriction sites. C. Northern blot showing examples of *BSD* mRNAs using different PPT lengths. The 14 T's in the parent reporter plasmid (giving 14 U's in the mRNA precursor, see panel F) were replaced by 9, 6, 3 or no T's. All results (except those for +BoxB with no PPT) were confirmed using at least three independent transformed populations. The methylene blue-stained rRNA shows the loading. WT are cells that do not have the plasmid. D. Northern blot for *PAC* mRNA. The WT lane shows a band labelled ** that cross-hybridised with the *PAC* probe. This appeared with varying intensities. E. *BSD* and *PAC* mRNAs from a +BoxB, 0-PPT clone which retained the *PAC* gene. F. Sequence between the *BSD* and *PAC* genes. The upper maps show the three different processing patterns: the partially processed bicistronic mRNA with no cleavage between the two open reading frames; the expected processing pattern; and the alternative processing pattern. The map of the intergenic region is expanded between the mRNA maps and the sequence. The sequence is colour-coded like the maps, and restriction sites are in lower case. The mapped processing sites are surrounded by boxes. The confirmed "expected" poly(A) sites are outlined in black, with the splice acceptor in blue. "Alternative" sites found with very short PPTs have the same colour code but are outlined with dotted lines. Start codons between the alternative and the wild-type PPT are underlined in green, and the wild-type *PAC* start codon in magenta.

poly(A) and splice site mapping by Northern blotting or reverse-transcription followed by PCR [6–9]. These experiments revealed that if the PPT that normally defined the splice site had been removed, an alternative sequence was used instead. Studies of splicing and polyadenylation sites by RNA sequencing (RNASeq) showed that many mRNAs use more than one AG splice acceptor site [23]. One of the early RNASeq studies revealed developmentally regulated splice sites [24]. Analysis of the RNASeq results also revealed that the lengths of PPTs vary from 9–40 nt; most are between 12 and 24 nt with a predominance of U over C. The distances between the preceding poly(A) sites and the PPT are usually 100–150 nt, and use of several different poly(A) sites is the norm. Siegel *et al.* [25] used a transiently transfected reporter, with RNA polymerase I transcription, to study the effects of PPT length and composition on *trans* splicing. They inserted various 17mers at the position of the PPT and then measured protein production from the resulting *trans* spliced RNA. Maximum efficiency was obtained with $(U)_{20}$. $(C)_{17}$ was inactive, but mixtures of Cs and Us were as good as Us alone. Insertions of purines, however, had clear deleterious effects.

Trypanosome precursor mRNAs, especially 3'-untranslated regions, contain numerous polypyrimidine tracts that are not normally used for *trans* splicing. Moreover, various studies have suggested that trypanosome mRNAs show differences in processing efficiency. Modelling of mRNA levels, based on high-throughput measurements, suggested that mRNA turnover rates and gene copy numbers could not account for all variations in mRNA abundance [26,27]. The authors suggested that if an mRNA precursor is inefficiently processed, some of the precursors will be destroyed in the nucleus by quality control mechanisms, resulting in low mRNA abundance.

In Opisthokonts, splice acceptor site choice is governed, at least in part, by specific RNA-binding proteins: Polypyrimidine tract binding proteins (PTBs) inhibit splicing [28], whereas some proteins with serine-arginine-rich (SR) domains define exons through binding downstream of splice acceptor sites [29]. Results from studies of selected trypanosome RNA-binding proteins and basal splicing factors indeed indicated that different factors control specific mRNAs. Possible regulators include DRBD3 and DRBD4, likely homologues of PTBs [30,31]; an HNRNPF/H homologue [32]; and various SR-domain proteins including TSR1 and its binding partner TSR1IP [33–35]. When expression of any of these proteins was reduced, some mRNAs increased and others decreased. The results suggested that depletion of the proteins affected the stabilities of some mRNAs, as well as processing of others. However, direct effects on mRNA processing were not actually demonstrated. The different effects presumably depend at least partially on whether the protein binds in the intron or in the body of the mature mRNA [32,36] but no binding sites in precursors have been identified.

One tool to study the effects of bound proteins on an RNA is the "tethering" assay [37,38]. In the most common arrangement, the protein of interest is expressed as a fusion with an RNA-binding domain, together with a reporter mRNA that includes the cognate recognition

sequence. The recognition sequence is generally inserted in the 3'-UTR because inclusion of RNA stem-loops, or binding of proteins, in the 5'-UTR can block translation [39,40]. One tethering system, derived from bacteriophage lambda, is a 22-residue peptide from the lambdaN protein, which binds with high affinity to the boxB sequence, a 19 nt stem-loop RNA [37]. The tethering assay has been used extensively, in various species, to study the functions of RNA-associated proteins in controlling mRNA decay and translation. The assay has various disadvantages: the function of the studied protein might be affected by the presence of the tag or by binding in an inappropriate place along the mRNA. Moreover, the fact that the binding is effected artificially via the lambdaN peptide can result in an inappropriate orientation or even incorrect folding of the protein. Nevertheless, predictions from tethering experiments have often been found to reflect the functions of the investigated proteins: examples from trypanosomes include various translation initiation factors [41,42], and RNA-binding proteins [43–45]. Tethering has also successfully been used to screen for post-transcriptional regulators [46–48]. However, tethering has, to our knowledge, not previously been used to investigate control of mRNA processing.

In this paper, we describe a reporter for mRNA processing, and its use in a screen for active PPTs. We also used tethering to demonstrate the effects of three mRNA processing factors (CPSF3, U2AF65, and SF1) and seven likely regulators (DRBD3 (PTB1), DRBD4 (PTB2), HNRNPH/F, TSR1, TSR1IP, RBSR1 and RBSR2) when tethered immediately upstream of the PPT. The results give useful insights into both the sequences required for active *trans* splice acceptor sites, and the functions of SR-domain proteins and PTB homologues in determining splice site choice.

## Methods

### Trypanosome culture and modification

The experiments in this study were carried out using monomorphic *T. brucei* Lister 427 bloodstream form parasites constitutively expressing the Tet repressor [49]. The parasites were routinely cultured at 37˚C in HMI-9 medium supplemented with 10% heat-inactivated fetal bovine serum (v/v), 1% (v/v) penicillin/streptomycin solution (Labochem international, Germany), 15 µM L-cysteine, and 0.2 mM β-mercaptoethanol in the presence of 5% $CO_2$ and 95% humidity. During proliferation, the cells were diluted to $1x10^5$ cells/ml and maintained between $0.2-2x10^6$ /ml. Cell densities were determined using a Neubauer chamber. For generation of stable cell lines, ~1-$2x10^7$ cells were transfected by electroporation with 10 µg of linearized plasmid at 1.5 kV on an AMAXA Nucleofector. Selection of new transfectants was done after addition of 5 µg/ml blasticidin (InvivoGen). Independent populations were obtained by serial dilution about 6 h after transfection. RNAi or tagged protein expression was induced using 100 ng/ml tetracycline.

### Plasmid construction

The primers and plasmids used are listed in S1 Table. Selected sequences are found in S1 Text and S1–S4 Files. The various components of pHD3180 (Fig 1B), which formed the basis of all subsequent experiments, were PCR-amplified using oligonucleotides that include appropriate restriction sites. A detailed description of the functions of the different DNA segments is at the beginning of the Results section. All reporter plasmids, with sequences, are available from the European Plasmid repository (https://www.plasmids.eu).

### PPT library preparation

Starting with pHD3180, 11 random nucleotides were inserted in place of the 14 Ts at the "PPT" position, by site directed mutagenesis using the primers in S1 Table. The resulting pool

of plasmids was de-salted by ethanol precipitation and transformed into TOP10 One Shot electrocompetent *E. coli* cells (ThermoFisher) by electroporation using 0.2 cm pre-chilled cuvettes. The electroporation was done at 2.5 kV, 25 μF, 200 Ω TC +/- 4.5 milliseconds. After transformation, the bacteria were plated on LB plates and incubated for 16–17 h at 37˚C, after which the colonies were picked and expanded overnight in 25 ml LB media and plasmids extracted. The plasmid library was transfected into *T. brucei* and selected with blasticidin as above. Surviving clones were then grown in 1x (200 μg/ml) or 5x (1 μg/ml) puromycin. Genomic DNA from the original population and the puromycin-selected parasites was extracted and the region targeting PPT amplified by PCR. The amplicons were run on a 3% agarose gel, and the products were cut, purified, and sent for sequencing (see below).

## RNA manipulation

RNA was prepared using Trizol according to the manufacturer's instructions. For Northern blots, 10 μg of total RNA was loaded per lane on formaldehyde denaturing gels, and blotted onto Nytran+. The membrane was stained with methylene blue and scanned before hybridization with [$^{32}$P]-labelled probes. Labelling with [α-$^{32}$P] dCTP was done using the Prime-It RmT Random Primer Labelling Kit (Agilent Technologies) according to the_ manufacturer's instructions. The signals were detected by phosphorimaging.

## DNA sequencing and data analysis

For DNA and RNA sequencing, libraries were prepared by David Ibberson at the CellNetworks Deep Sequencing Core Facility at the University of Heidelberg, followed by sequencing at EMBL.

For the PPT analysis, the relevant integrated plasmid sequence was amplified (S1 Table) from total population DNA and then sequenced (S1 Table). Raw paired Illumina (Miseq) reads were merged using the fastq-join tool with default parameters (version 1.3.1). A custom bash code was used to subset PPT sequences by searching for the flanking sequences (GGCGAAATCTAGA and AGATCTACTTC). The number of unique PPT sequences, their lengths and composition were counted and used for further visualization in R (version 3.6.3; R Core Team, 2022) on Ubuntu 20.04.2 LTS. The dplyr (version 1.0.8) [50], tidyverse [51] and ggplot2 [52] packages were applied in data manipulation and plotting in R. Sequences that were depleted after both 1x and 5x selection were analyzed using WebLogo [53].

Total RNA was either poly(A)-selected using the standard Illumina kit, or subjected to rRNA depletion using oligonucleotides and RNase H [54]. RNASeq data were aligned to the TREU927 and Lister 427 (2018) genomes using a custom pipeline that includes Bowtie2 [55,56] and then statistically analysed using a DeSeq2-based pipeline [57,58]. Subsequent analyses were done mainly in Microsoft Excel.

## Drug sensitivity tests

To measure the $EC_{50}$ for puromycin, the drug was serially diluted in 100 μl culture medium in 96 well opaque plates to give concentrations between 50 ng/ml and 26 mg/ml. Trypanosomes were added at a final density of $2 \times 10^4$/ml (100 μl) and incubated for 48 h. After incubation, 20 μl of 0.49 mM Resazurin sodium salt in PBS was added to each well and plates were incubated for a further 24 h. Plates were read on a BMG FLUOstar OPTIMA microplate reader (BMG Labtech GmbH, Germany) with $\lambda_{excitation}$ = 544 nm and $\lambda_{emission}$ = 590 nm, to assess the number of surviving viable cells [59,60]. GraphPad Prism 6 was used to calculate $EC_{50}$. The percentage growth inhibition was plotted against the $\log_{10}$ puromycin concentration and analysed in sigmoidal dose response, variable slope mode.

## Mass spectrometry

Proteins associated with RBSR1 and RBSR2 were identified exactly as described in [61]. Briefly, RBSR1-TAP, RBSR2-TAP and TAP-GFP were purified from cell extracts over IgG beads, then released using His-tagged TEV protease. The protease was subsequently removed using a nickel column. The purified proteins were then run on a polyacrylamide gel and subjected to mass spectrometry. Data were analyzed using PERSEUS [62].

## Results

### Construction of the reporter

The reporter used for our studies is shown in Fig 1B(a). After restriction site cleavage, it integrates into the tubulin gene repeat, replacing an alpha-tubulin open reading frame. This will result in transcription of the reporter by RNA polymerase II. The first open reading frame, encoding blasticidin S deaminase (BSD), is preceded by the native alpha-tubulin 5'-UTR and upstream splicing signals. After the *BSD* gene comes a truncated version of the 3'-UTR from the actin-B (*ACT*) gene, followed by the downstream intergenic region (IGR). The polyadenylation site is fortuitously marked by a *Sal* I site. The PPT from the actin intergenic region is separated from the IGR by two unique restriction sites (*Xho* I, *Xba* I), and is followed by a unique *Bgl* II site. Next, a fragment of the *EP* procyclin 5'-UTR, with a few additional nucleotides, precedes an *Spe* I site, followed by an open reading frame encoding puromycin acetyltransferase (PAC), and finally, the alpha-tubulin 3'-UTR. Two versions of this plasmid are available, pHD3180 and pHD3259: they differ only in that pHD3259 has a shorter segment of the alpha-tubulin locus upstream of *BSD*. After transcription of the reporter plasmids, the *BSD* mRNA has an alpha-tubulin 5'-UTR and a truncated *ACT* 3'-UTR, while the *PAC* mRNA has the *EP1* 5'-UTR (plus additional sequence) and the alpha-tubulin 3'-UTR.

All experiments were done using Lister 427 strain bloodstream-form trypanosomes expressing the Tet repressor [49]. These cells are ideal for high-throughput analyses but cannot differentiate into replication-competent procyclic forms. It is conceivable that some results might differ in differentiation-competent cells.

### The effect of changing the PPT on mRNA processing

Throughout this study, we used Northern blotting to detect mRNAs from cells with the integrated plasmids, because it reveals mRNA lengths. We used radioactive probes for *BSD* and *PAC* to attain maximum sensitivity, and because they give quantitative results (over a limited range). Unlike reverse transcription and quantitative PCR, Northern blots require no assumptions about the products to be obtained. Results for the parent plasmid are in the far left-hand lanes of the blots shown in Fig 1C and 1D (labelled -boxB, 14 Us). The precursor transcript has 14 consecutive U's in the PPT position, corresponding to the sequence originally published for the actin locus [63]. In Fig 1F, the top map shows a bicistronic mRNA, which has been processed using only the tubulin signals (see below), and the next map down shows the products expected from the original construct, with $(U)_{14}$. Below the maps in Fig 1F, we show the sequence between the *BSD* and *PAC* open reading frames. The fully processed *BSD* and *PAC* mRNAs were both expected to be about 850 nt long, assuming use of the endogenous tubulin signals [64] for *trans* splicing of *BSD* and polyadenylation of *PAC*, and a mean poly(A) tail length of 60 nt. The expected mRNAs were indeed seen (Fig 1C and 1D). Use of the anticipated sites was confirmed by reverse transcription, PCR and sequencing. We found a single splice site (dark blue box) for *PAC*, and two different poly(A) sites (black boxes) for *BSD*. Use of several different polyadenylation sites is normal for trypanosome mRNAs.

In addition to the expected or "wt" mRNAs, an mRNA that hybridized with both open reading frame probes (labelled "bi" in all subsequent Figures) was always present. This suggested that processing between the *PAC* and *BSD* mRNAs was inefficient. From the length of this mRNA, we assume that it is *trans* spliced upstream of *BSD* and polyadenylated downstream of *PAC* (Fig 1F, top map), but we have not verified this experimentally. We also do not know whether the bicistronic mRNA is exported to the cytoplasm. Similar bicistronic tubulin mRNAs, which also result from incomplete processing, are usually substrates for degradation by the RNA exosome, most of which is in the nucleus [65]. RNAs containing two or more open reading frames also accumulate after inhibition of splicing or polyadenylation [10,11,66], and can be exported, at least partially [67]. Relative to the presumed bicistronic mRNA, the level of the *PAC* mRNA appeared higher than for the *BSD* mRNA. This might have been due to less efficient 5' processing of the *BSD* mRNA, or faster polyadenylation of the PAC mRNA, but since the former two processes are coupled, we think that the difference is more likely to have been caused by different mRNA half-lives, becuase the two mRNAs have different 3'-UTRs. Full-length *ACT* mRNA has a half-life of about 20 min, whereas the tubulin mRNA has a half-life of at least one hour [26].

Next, we examined the effect of shortening the PPT. Reducing it to $(U)_9$ made no difference to the splicing pattern (Fig 1C and 1D, lane labelled -boxB, 9), but further shortening caused the appearance of a weak *PAC* band migrating at about 1.1 kb and a *BSD* mRNA that was about 100nt shorter than the expected size (Fig 1C and 1D, lanes "-boxB, 6, 3, 0", bands labelled "alt" for "alternative"). To locate the splicing and polyadenylation sites of the alternative *PAC* and *BSD* mRNAs, we again amplified them and sequenced the products. Results are illustrated in Fig 1F. With the shortened PPTs, the *BSD* poly(A) site was 20 nt downstream of the *BSD* cassette (black dotted box), giving an mRNA that was predicted to be 200 nt shorter than the original. The alternative *PAC* mRNA splice site was correspondingly 104 nt downstream of the new poly(A) site, *i.e.*, 200 nt upstream of the original one (blue dotted box). The longest PPT upstream of the new splice site is UUUUCCUC but this is only 43 nt downstream of the new poly(A) site, which would be unusually close. Other pyrimidines in more appropriate positions are highlighted in purple. The new *PAC* mRNA has an extended 5'-UTR which contains three initiation codons (underlined), with downstream open reading frames of 8, 18 and 23 codons and terminating upstream of the *PAC* initiation codon. We would expect the presence of these "upstream open reading frames" to impair translation of the *PAC* gene severely.

Knowing the processing sites, it might have been possible to devise a real-time PCR assay at least for the wild-type spliced *PAC* and the truncated *BSD* poly(A) site. However, design of specific primers for quantification of the other products seemed likely to be difficult, due to the heterogeneous poly(A) site use, and the AT-richness and low complexity of the intergenic region. Moreover, we were worried that by using real-time PCR we might miss additional alternatively processed products. We therefore continued to use Northern blotting for our studies.

We next looked at the effects of PPT changes on puromycin resistance, measuring survival after two days of treatment using a live-cell fluorescence assay. A typical set of curves is shown in Fig 2A and results for three replicates are in Fig 2C. As expected, puromycin resistance depended on production of mRNA with the short 5'-UTR. Decreasing the length of the PPT from 14 to 9 nt had no effect on puromycin resistance and with no PPT the $EC_{50}$ was two orders of magnitude lower. Cells with $(U)_6$ were marginally more resistant than those with $(U)_3$, but they still had an $EC_{50}$ that was ten times lower than the $(U)_{14}$ control value. These results showed that the vector can be used not only to test the effects of specific intergenic regions, PPTs, or 5'-UTRs on gene expression, but also for untargeted screens.

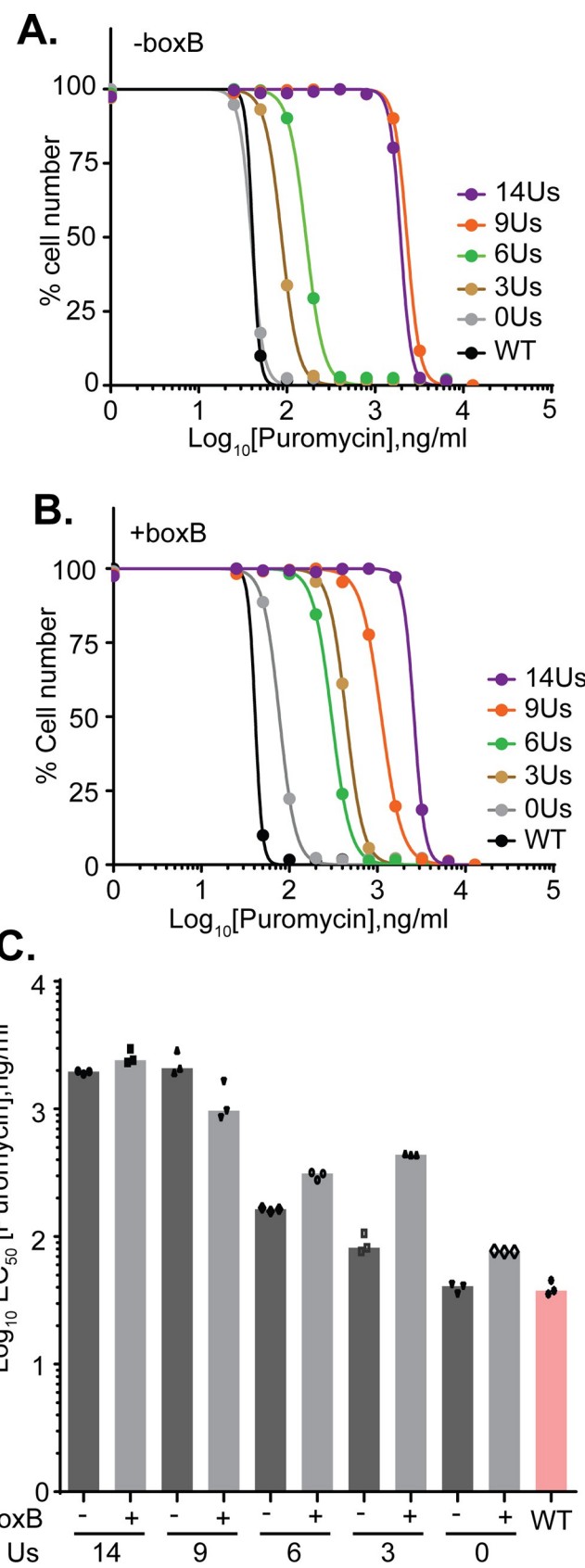

**Fig 2. Effect of different PPT lengths on puromycin sensitivity.** A. Puromycin sensitivity curves for trypanosomes containing pHD3180 (no boxB) with different PPT lengths. Numbers of viable cells remaining after 48 h (as measured using rezazurin) relative to untreated cells are shown on the y-axis, and the puromycin concentration on the x-axis. B. As (A) but for trypanosomes containing pHD3186 (with boxB) with different PPT lengths. C. Quantitation of three technical replicates for clones shown in Fig 1. Additional clones gave similar results.

## Addition of a boxB sequence upstream of the PPT promotes splicing

One of our aims was to investigate the effects of potential splicing regulators on our reporter. To enable this, we incorporated two copies of boxB immediately upstream of the PPT. We then tested the effects on mRNA processing (Fig 1C–1E) and puromycin resistance (Fig 2B and 2C). The presence of 2xboxB had no effect for $(U)_{14}$ but slightly inhibited processing with $(U)_9$. In contrast, insertion of 2xboxB upstream of $(U)_3$ yielded results that were similar to those from $(U)_6$ with boxB, suggesting that with $(U)_3$, 2xboxB was stimulating processing.

With no PPT at all and 2xboxB, Northern blots combined with PCR using genomic DNA showed that four of five clones tested had deleted the *PAC* gene entirely, giving just the normal *BSD* transcript (# in Fig 1C and 1D). Only one clone had retained the *PAC* gene, and this had only the bicistronic *BSD-PAC* mRNA (Fig 1E). This suggests that 2xboxB requires the 3 Us in order to give "correct' splicing, and has no activity by itself. For quantification of puromycin resistance, this clone was used. The *PAC* gene deletion also occurred with two of the five clones with no PPT and without boxB, and in one clone with $(U)_3$ and no boxB. Since all clones were selected with blasticidin, this suggests that the bicistronic RNA alone gives very poor blasticidin resistance, consistent with either retention in the nucleus, or repression of translation caused by the *PAC* sequence in the 3'-UTR.

## Different 5' regions had small effects on mRNA yields

In previous work, we measured both the abundances and half-lives of most mRNAs in *T. brucei* bloodstream forms [26]. We also generated a mathematical model of gene expression. To do this we assumed that both the transcription and processing rates were the same for all mRNAs, which would mean that abundances were mainly determined by mRNA half-lives, To obtain the best fit between the predicted and measured mRNA abundances we had to include a deleterious effect of mRNA length [26,27]. For a minority of mRNAs, the predictions of mRNA abundances from the model did not match measured mRNA abundances, suggesting that for these mRNAs, either the measurements or the assumptions were incorrect.

One possible explanation for the discrepancies was that mRNAs differed in their processing efficiencies. To investigate this, we used our new reporter vector to measure the effects of different IGR-PPT-5'-UTR segments on mRNA abundance. The region between two actin genes (*ACT*) served as a control. We chose five additional mRNAs according to the match between predicted and measured mRNA abundances [27]. The mRNA for Tb927.7.3940, encoding a mitochondrial carrier protein (MCP16) was expected to have an average *trans* splicing efficiency (i.e. the abundance matched the prediction). Tb927.7.940 mRNA, encoding a protein of unknown function, and Tb927.11.13780 mRNA, encoding profilin (PROF), were expected to be spliced faster than average (abundances approximately twice the predictions). In contrast, mRNAs from Tb927.10.7420, encoding bromodomain factor 2 (BDF2), and Tb927.11.3270, encoding squaline monooxygenase (SQM), were expected to be processed slower than average (abundances four-to-five times lower than predicted). We verified manually, by examining transcriptomes in TritrypDB (https://tritrypdb.org/tritrypdb/app), that the mRNA lengths that had been used for the model predictions were correct.

For each gene, we amplified the region from the polyadenylation site of the preceding gene to the nucleotide immediately before the start codon, thus including the upstream IGR with splicing signals as well as the 5'-UTR (S1 Text). We used the resulting fragments to replace the region between the *Sal* I and *Spe* I sites in our starting plasmid pHD3180 (Fig 3A). The constructs were therefore designed such that the 3'-UTR of the *BSD* mRNA should remain unchanged, but the *PAC* mRNAs would have the 5'-UTR from the gene under study. For *ACTA*, both the *PAC* and *BSD* mRNAs were the same size as before, as expected (Fig 3B). Processing was notably more efficient than it had been for the starting plasmid, since no bicistronic mRNAs were visible. This suggests that the patchwork of different sequences and/or the added restriction sites in pHD3180 had impaired processing efficiency. Although this was surprising, it was also serendipitous because it meant that pHD3180 could be used to detect both increases and decreases in processing efficiency.

All of the other *PAC* mRNAs had the expected lengths: for *PROF*, *BDF2*, and Tb927.7.940 similar to *ACT*, but longer for *MCP16* and *SQM* owing to their longer 5'-UTRs (S1 Text and Fig 3B). The *BSD* mRNAs also all had the same lengths, as planned, except in the case of Tb927.7.940. For this last plasmid, the *BSD* mRNA was too long (Fig 3B); this was presumably because our cloned fragment accidentally included about 140 nt of the preceding 3'-UTR, which would cause a 3' extension of the *BSD* mRNA. Changing the intergenic region had very little effect on the abundances of the *PAC* or *BSD* mRNAs: all of the different upstream regions gave 20–40% less mRNA than the actin controls (Fig 3C). Thus, if these mRNAs really are spliced with different efficiencies, the sequences we used do not contain the requisite information. One possibility is that the beginning of the coding region, which was not included in our test constructs, might affect processing. Another possibility is that the high-throughput mRNA half-life measurements that were used to predict the abundances of the mRNAs [26] were incorrect.

## A screen for splicing signals

Next, we tested whether the original reporter plasmid could be used for high-throughput screening. To do this, we made a library of plasmids containing different 11mers in the PPT position. The RNA sequence that results is ucuagaNNNNNNNNNNNNAagaucu (restriction sites in lower case), so that the first available splice acceptor site (underlined) is immediately downstream of the 11mer. A second potential AG acceptor is located 37 nt downstream, 10 nt upstream of the AUG. The plasmids were transfected into trypanosomes and selected with blasticidin. (We note that since the bicistronic mRNA seems to give poor blasticidin resistance (Fig 1C), selection with blasticidin might, by itself, select against PPT-region sequences that prevent processing.) The resulting population was grown in puromycin. DNA was prepared, and the "PPT" regions from different populations were amplified using upstream and downstream primers (S1 Table) and then sequenced. An initial trial revealed selection of specific sequences at 0.2 μg/ml or 1 μg/ml puromycin (S2 Table), and no survival at 2 μg/ml. We therefore repeated the experiment with three independent replicates, first with 0.2 μg/ml puromycin (the normal concentration, 1x in Fig 4), which initially inhibited growth before recovery of the population, and then 1 μg/ml drug (5x in Fig 4), which did not detectably retard replication. Sequencing of the amplicons yielded 2.5–3.3 million reads per population, with up to 20,731 unique sequences in a replicate and coverage from 1x to 790,634x. Selection for 11mers only revealed 30,229 different 11mers, of which 28,248 were present as less than ten reads in the initial population, and 11,135 were completely absent. To find sequences that were lost during selection, we concentrated on those that were initially present as at least ten reads, and for which the median normalized values after puromycin selection were at least five-fold lower.

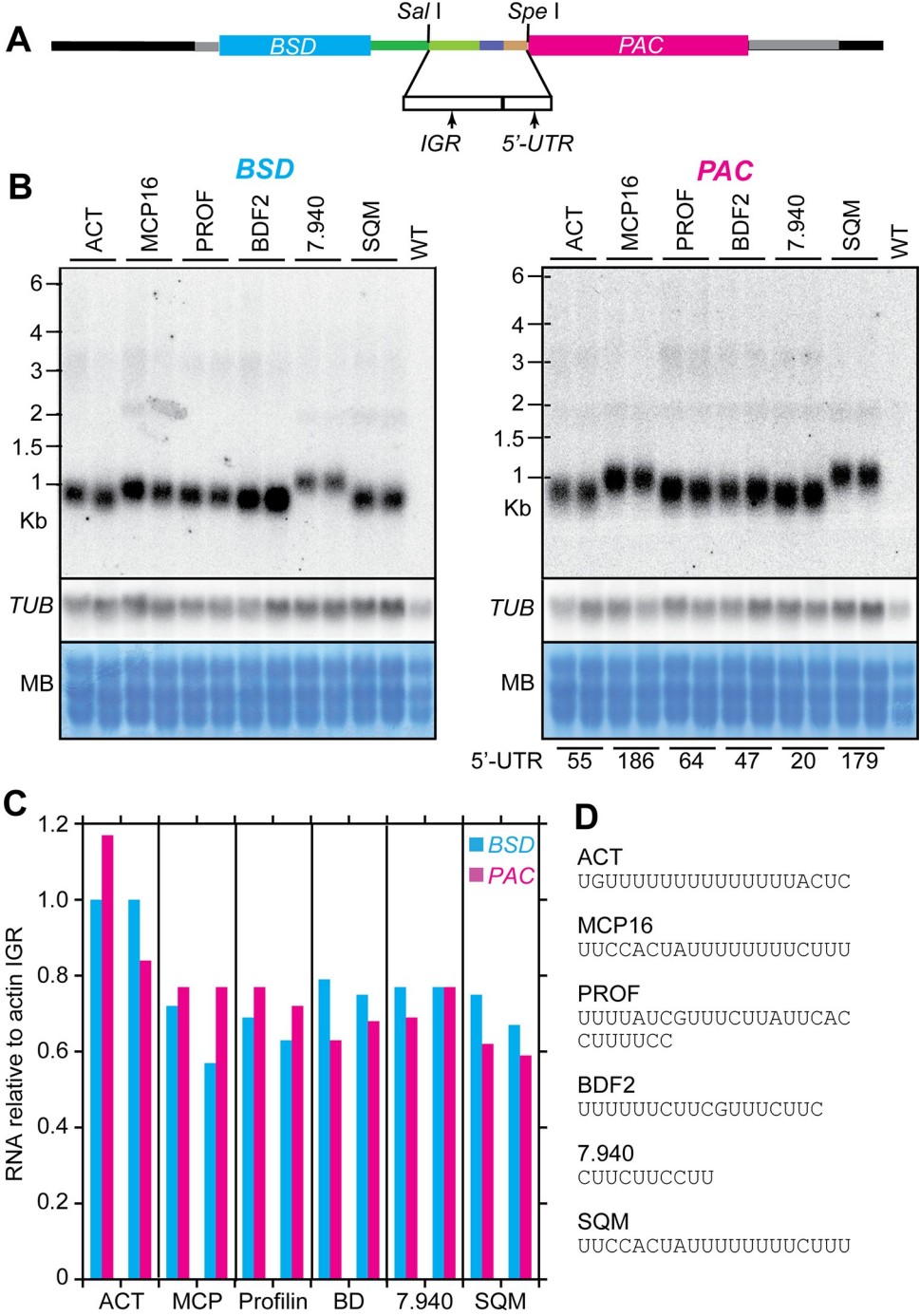

**Fig 3. Different intergenic regions and 5'-UTRs.** A. Cloning scheme showing positions of the tested sequences. B. Northern blots as in Fig 1C and 1D. The blot was hybridized with a beta-tubulin probe for normalization. Results of two different populations were used. The predicted 5'-UTR lengths are noted below the lanes. C. Amounts of mRNA, obtained by scanning the Northern blot images. Amounts were calculated for two different Northern blots for each population and normalized to the tubulin signal. D. Polypyrimidine tracts upstream of the splice sites. The full sequences are shown in S1 Text.

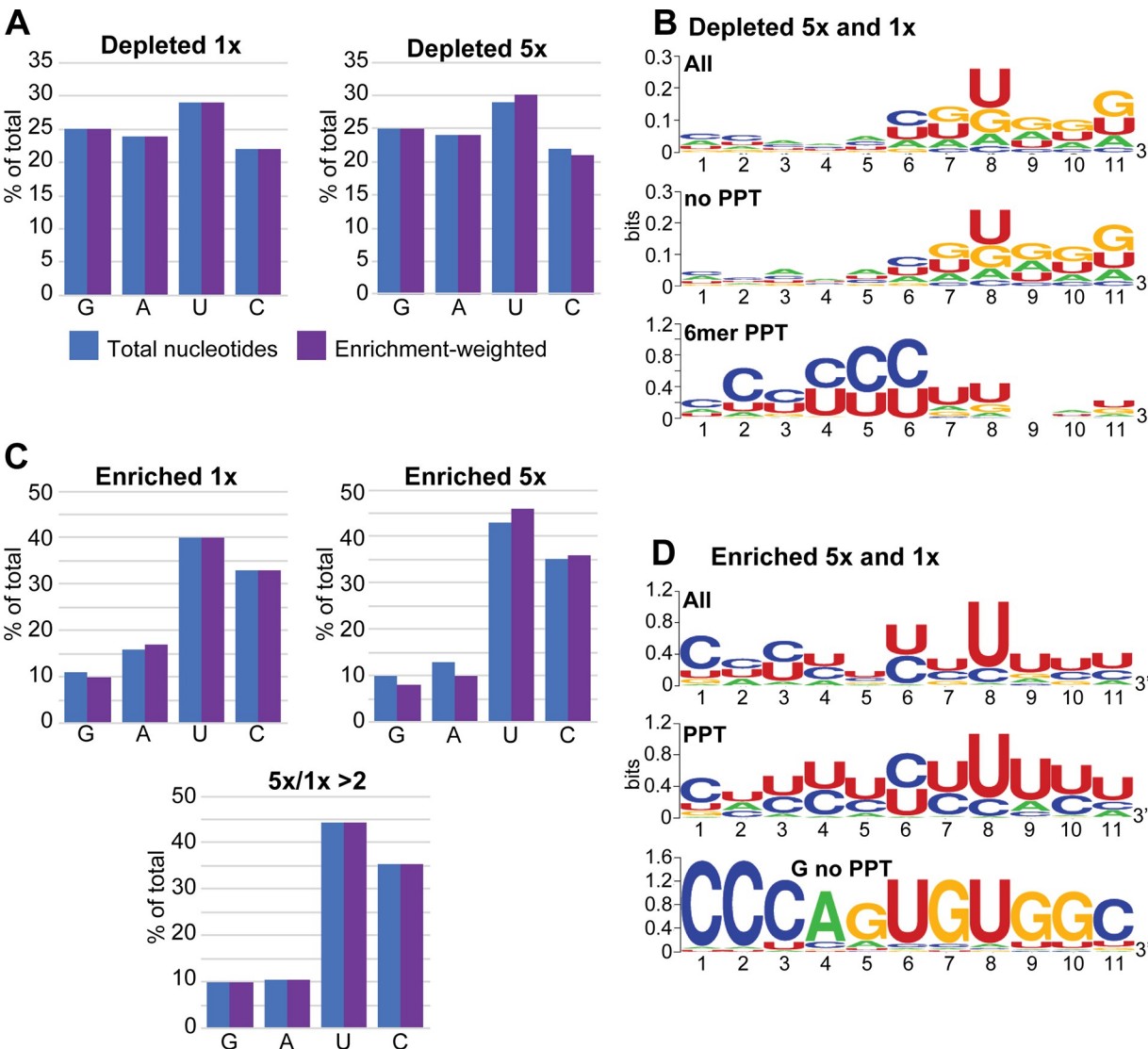

**Fig 4. 11mer sequences that affect reporter expression.** Trypanosomes carrying the reporter (no boxB) with random 11mers in the PPT position were first selected with blasticidin and then with puromycin. Inserts of the starting and selected populations were amplified and sequenced. A. Nucleotide compositions of depleted sequences. We selected sequences that were present at least ten times in the starting population. To determine enrichment in the starting population, the normalized number of reads (reads per million reads, RPM) was then divided by the (median RPM +1) for three populations selected with 0.2 μg/ml or 1 μg/ml puromycin (Depleted 1x and Depleted 5x, respectively). Sequences for which the enrichment in the starting population was at least five-fold were selected. The blue bars show the base compositions if every selected sequence was counted once. The purple bars show the base compositions after each sequence was weighted according to the degree of selection (multiplied by the enrichment factor). B. Sequences that were depleted after both 1x and 5x selection were analyzed using WebLogo [53]. The top panel shows all sequences, the middle panel those that lack six contiguous pyrimidines, and the bottom panel those that had six contiguous pyrimidines. C. Base compositions of sequences that were enriched at least five-fold after growth in 1x or 5x puromycin. Details are as in panel A. The bottom graph shows results for sequences with a median enrichment in 5x puromycin that was at least two-fold higher than in 1x puromycin. D. Consensus sequences for 11mers that were at least five-fold more enriched with 1x puromycin, and either as enriched, or even more enriched, with 5x puromycin. Results for all 194 sequences are shown at the top, those for 37 sequences with at least one G and no 6mer PPT are at the bottom, and those for the majority that remained are in the middle.

This yielded 1,177 11mers (Fig 4A) with very little base preference at any position, apart from a very weak preference for G and A towards the 3'-end (Fig 4B). Intriguingly, however, 120 of them contained a PPT of at least 6 nt. Separate examination of this subset revealed that the PPTs were concentrated towards the 5'-end of the 11mer (Fig 4B).

After selection in 1 µg/ml puromycin, the population included 230 different 11mers with median enrichment of at least five-fold. Further selection with 5 µg/ml gave 266 sequences. In both cases, there was selection for C and U residues and strong loss of G (Fig 4C). 194 sequences were five-fold enriched under both conditions, and at least as abundant with 5 µg/ml as with 1 µg/ml. These showed a preference for U or C in all 11 positions, with particularly strong preference for U at position -4 relative to the AG dinucleotide (Fig 4D). 80 of the 11mers included at least ten pyrimidines and 126 included at least nine, and this preference was strongest for sequences that increased in abundance under higher selective pressure (Fig 4C).

Strangely, 45 of the selected sequences lacked six consecutive pyrimidines, and included one or more G residue. These had a strong consensus sequence, "CCCAGUGUGGC" (Fig 4D, panel labelled "G no PPT"), which itself was more than 20x selected. This sequence is completely absent from the reference TREU927 genome. We also looked for some of the related sequences in the 50 nt upstream of all (mapped) splice sites and found only very few examples, all of them upstream of a consensus U/C-rich PPT. To find out whether "CCCAGU-GUGGC" was active instead of a PPT, we tested it individually, but cells containing the plasmid were as susceptible to puromycin as the wild-type. We therefore cannot explain why this and related sequences were selected in our screen. Nevertheless, the function of this sequence would probably warrant further investigation.

## Tethering of CPSF3 or SF1 enhances splicing

Next, we tested whether the BoxB-containing reporter could be used to assay the functions of known processing factors, or potential regulators. To do this, we expressed selected proteins of interest as fusions, with the lambdaN peptide at the N-terminus, and two myc tags at the C-terminus. Fig 5A shows schematically the precursor RNA with the bound lambdaN fusion protein. Initially, we used the reporter with $(U)_3$ at the PPT position, aiming to identify factors that stimulated processing. All results from the tethering must be interpreted with caution because protein function might be impaired or altered by the presence of the tags. Also, the tethered protein might have an inappropriate orientation or position relative to the RNA backbone or the splice site. It was therefore important to check whether proteins of known function had the expected effects.

As a negative control, we expressed lambdaN-GFP-2myc, which had no effect on processing or puromycin resistance (Fig 5B, replicates and Western blots in S1 Fig). Next, we tethered CPSF3, the cleavage and polyadenylation factor, and saw strongly enhanced processing at the "correct" sites and a more than ten-fold increase in puromycin resistance (Figs 5C and S2). The correct processing in this case was rather surprising since we had expected this enzyme to cleave near the tethering site, but the increase in resistance showed that screening would be possible. Moreover, tethering of the splicing factor SF1 also enhanced processing (Figs 5D and S2). In contrast, tethering of the putative homologue of U2AF65 shunted some processing to the upstream positions, with marginally reduced puromycin resistance (Figs 5E and S3).

## Tethering of potential splicing regulators

Next, we investigated nuclear proteins that had previously been shown to influence mRNA levels. DRBD3 and DRBD4 are possible homologues of Opisthokont PTBs. This identification is based on the locations and sequences of RNA-recognition motifs, and is described in detail in [30]. Tethering of either protein strongly inhibited processing between *BSD* and *PAC*, with a possible increase in the abundance of the bicistron and increased puromycin susceptibility (Figs 6A and 6B and S4). In contrast, tethering of HNRNPF/H had no effect (Figs 6C and S1). Tethering of the SR-domain protein TSR1 and its interaction partner TSR1IP, in contrast,

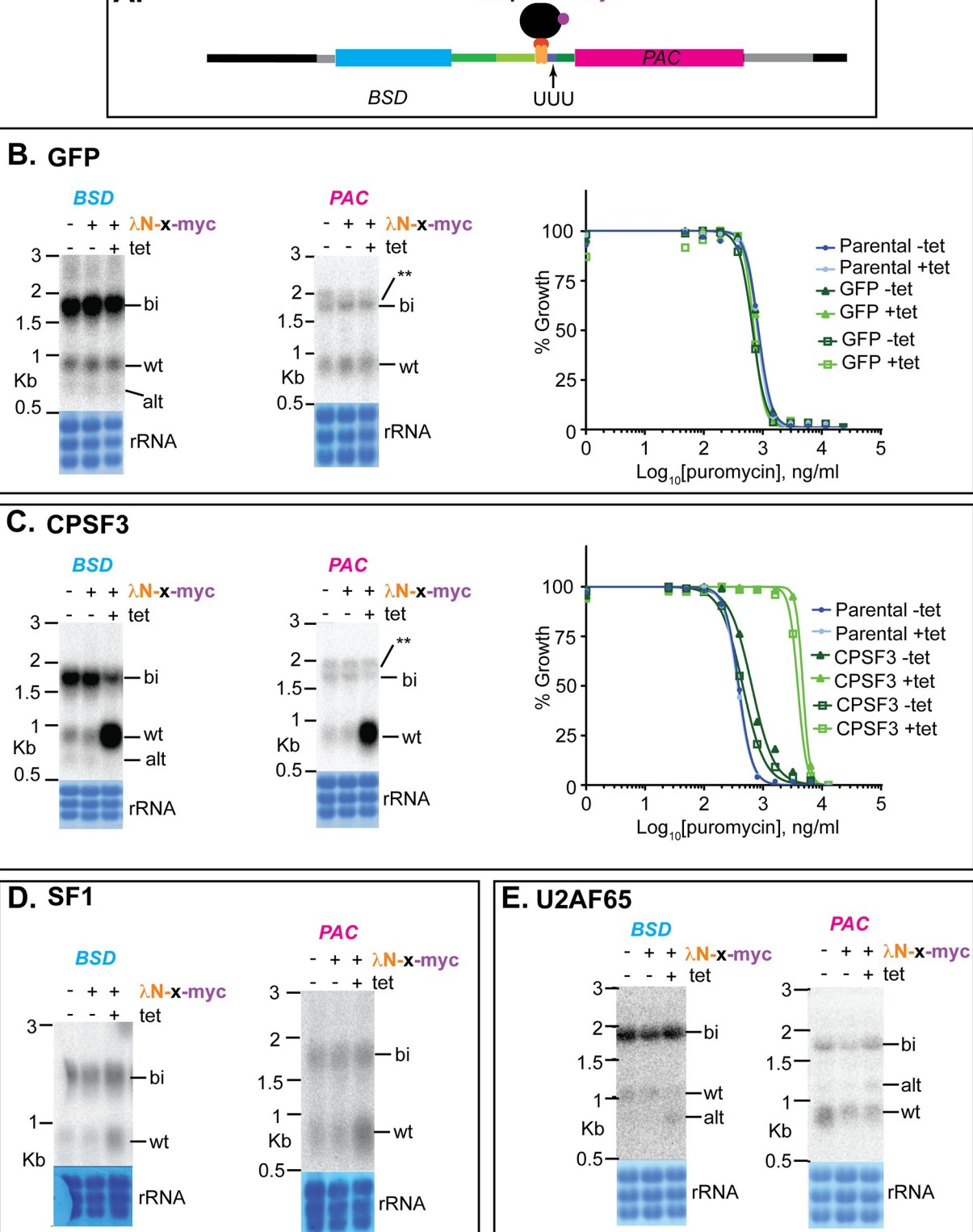

**Fig 5. Effects of tethering GFP, CPSF3, SF1 or U2AF65 to the boxB-(U)₃ reporter.** A. Cartoon showing the precursor before processing, with tethered protein. B. Tethering of lambdaN-GFP-myc. In this and all subsequent similar panels, a Northern blot is shown for cells with no inducible lambdaN protein, then one clone with and without tetracycline. Results for both *BSD* and *PAC* are illustrated, with the stained rRNA on the membrane below. The susceptibility of two different clones (with different symbols) to puromycin is shown on the right. Other labels are

as in Fig 1C. Results for additional clones and biological replicates are shown in S1–S3 Figs. C. Tethering of lambdaN-CPSF3. D. Tethering of lambdaN-SF1. E. Tethering of lambdaN-U2AF65.

resulted in use of the alternative sites (Figs 7A and 7B and S5), again with decreased puromycin resistance. We also investigated two additional SR-domain proteins: RBSR1, which is mainly in the nucleus, and RBSR2, which is present in both nucleus and cytoplasm [68]. Each of these had the same effect as TSR1 (Figs 7C and 7D and S6).

So far, the experiments had employed a reporter with an extremely short PPT. We therefore asked whether the potential regulators would also affect splicing of a precursor with stronger PPTs. Using the 2xboxB-(U)$_9$ reporter, HNRNPF/H and U2AF65 again had no effect. DRBD3 and DRBD4 inhibited splicing, and the SR-domain proteins again caused use of the alternative sites (Figs 8 and S7–S9). With 2xboxB-(U)$_{14}$, effects were similar except that for RBSR1 and RBSR2, splicing was inhibited and the alternatively processed products were not detected (Figs 8 and S7–S9).

## The protein associations of RBSR1 and RBSR2

Tethering of RBSR1 and RBSR2 had suggested that they both might have functions in defining splice sites. Since neither had previously been investigated in *T. brucei*, we decided to follow up the tethering result. First, we characterized their protein associations. Cell lines were made that expressed C-terminally tandem affinity-tagged RBSR1 or RBSR2 from the endogenous locus. After purification over IgG beads, the tagged proteins were released using His-tagged TEV protease, which was subsequently removed using a nickel column. The extracts were then analyzed by mass spectrometry, using inducibly-expressed, similarly tagged GFP as the control. RBSR1 associated with 131 proteins, including 22 components of the splicing machinery (Fig 9A and S3 Table), five components of the PRP19 complex as well as U2 and U5 snRNP components (S4 Table). Interestingly, the CFII subunit of the polyadenylation complex was also present. Although RBSR1 is predominantly in the nucleus, it was also associated with ribosomes and with one of the cap-binding translation initiation complexes, EIF4E4/EIF4G3. There was little overlap with proteins that co-immunoprecipitated with GFP-tagged *Trypanosoma cruzi* RBSR1, although that study also identified TRRM1 and some ribosomal proteins [69]. Our study was probably more sensitive since we used quantitative proteomics.

The RBSR2 purification yielded fewer (56) specifically associated proteins than RBSR1 (Fig 9B and S3 Table), perhaps because RBSR2 is (according to quantitative mass spectrometry) about four-fold less abundant [70] (S4 Table). RBSR2 was not clearly associated with the mRNA processing machinery but pulled down four proteins from the basal body, and again various ribosomal proteins (Fig 9B and S3 Table). A comparison of the two proteomes (Fig 9C) revealed that although TSR1 was present in both, it was more enriched with RBSR2. Vault proteins also copurified with both RBSR1 and RBSR2, but more with RBSR2, and RBSR1 pulled down many more (potentially) regulatory RNA-binding proteins, including DRBD3.

## Effects of RBSR1 and RBSR2 depletion on growth and the transcriptome

To examine the roles of RBSR1 and RBSR2 in mRNA metabolism, we depleted them by inducible RNA interference (RNAi) and examined the transcriptomes, starting with cells expressing the corresponding TAP-tagged protein. The lines with best depletion of RBSR1 had, without induction, roughly half as much RBSR1 protein as the starting (tagged) cell line. After RNAi induction, the level decreased to about 20% (Figs 10A and S10). The cell lines grew almost normally and RNAi induction had no effect (Fig 10A). For RBSR2, RNAi decreased the protein

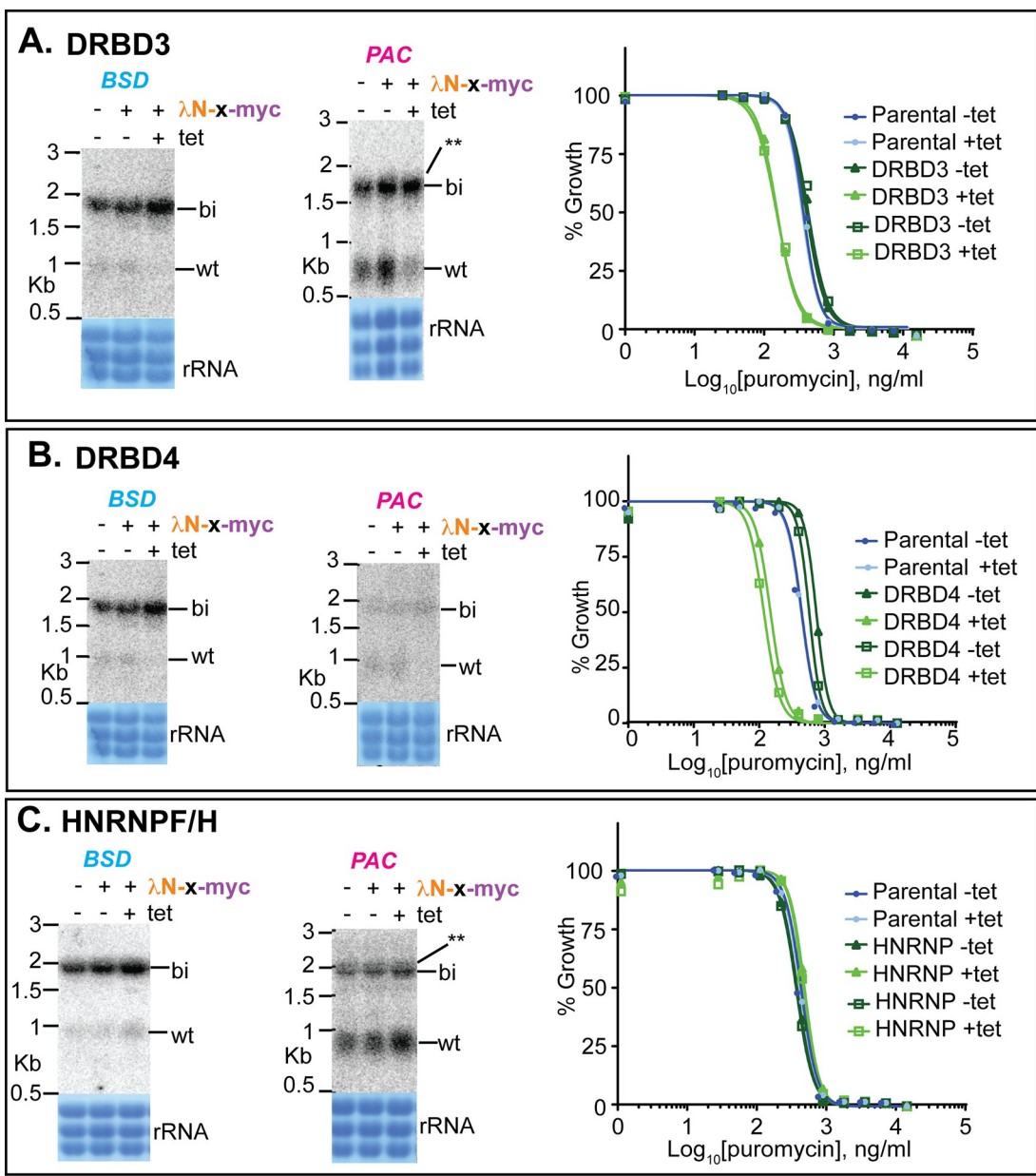

**Fig 6.** Effects of tethering (A) DRBD3, (B) DRBD4 or (C) HNRNPF/H to the boxB-(U)$_3$ reporter. Details are as for Fig 5. See also S1 and S4 Figs.

level to around 11%, but there was also no effect on growth (Fig 10B). We compared the transcriptomes of two independent *RBSR1* RNAi lines and three independent *RBSR2* RNAi lines (Figs 10A and S10) with three replicates from the RBSR2-TAP starting line (chosen because the TAP tag alone had no effect on growth). For both RBSR1 and RBSR2, RNAi induction had no effect on the transcriptome (S8 Fig and S5 and S6 Tables). All of the RBSR RNAi lines showed accumulation of the RNAi-mediating RNA, which, from examination of the reads (E-MTAB-11627 and E-MTAB-11648), was *trans* spliced on the "sense" strand.

We pooled the RNASeq results with and without tetracycline, since it made no difference, and compared the transcriptomes of the two lines with the starting (TAP-tagged) line for

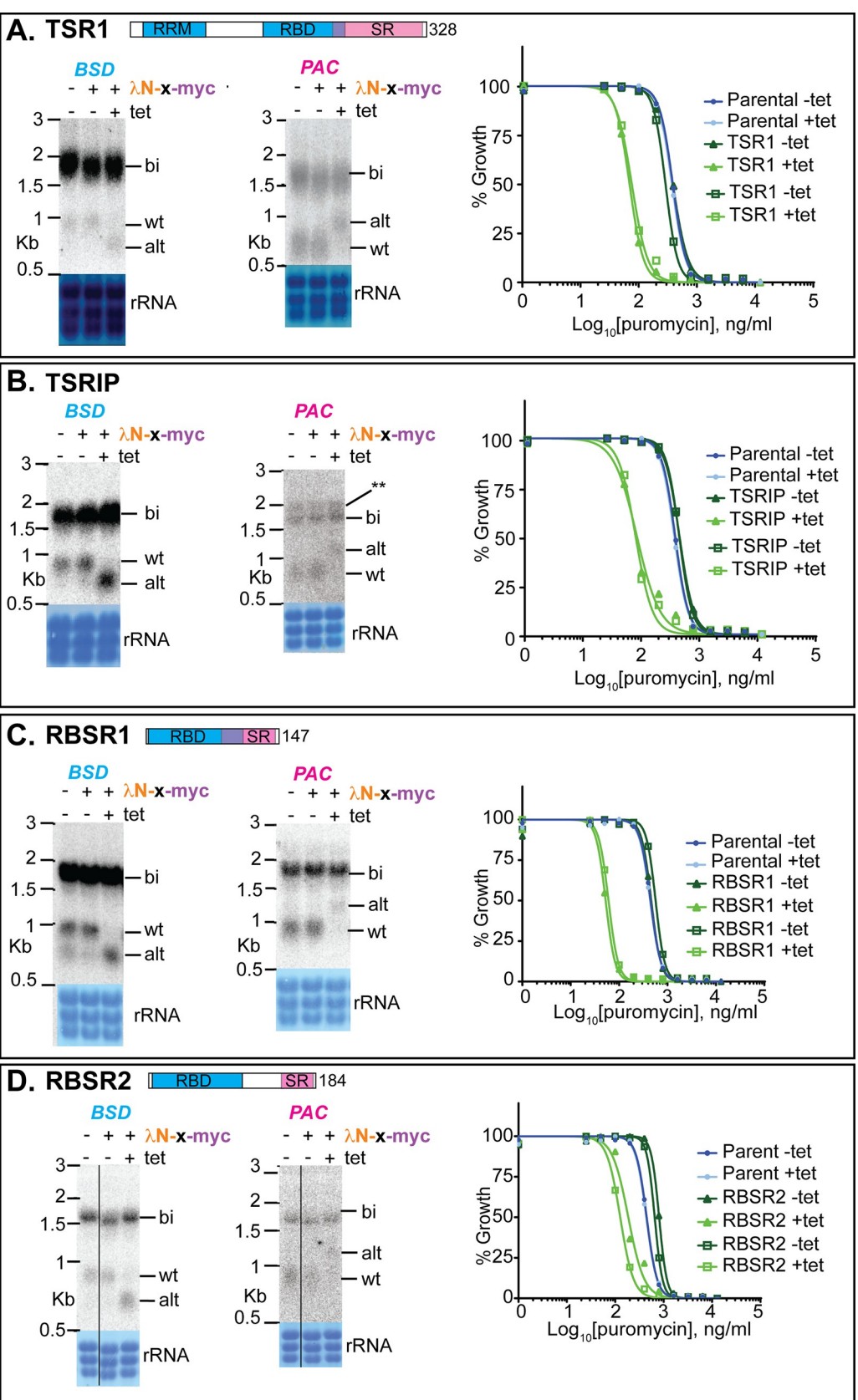

**Fig 7. Effects of tethering SR-domain proteins to the boxB-(U)₃ reporter.** Details are as for Fig 5. See also S5 and S6 Figs. A. Tethering of TSR1. A cartoon of the protein sequence is shown at the top, with the total length in residues indicated. RRM (RNA recognition motif) domain: IPR000504. The precise boundaries of the serine-arginine-rich domain are unclear because there is no standard definition. The region shown has 50%R, 38%S, including (RS)₈. B. Tethering of TSR1IP, which interacts with TSR1 [33–35]. TSR1IP has no RNA-binding domain, C. Tethering of RBSR1. The RNA binding domain is SSF54928, and the SR-domain shown is 50%R, 32%S including two copies of (RS)₃. C. Tethering of RBSR2. The RNA binding domain is IPR035979. THE SR-domain illustrated is 27%R, 51%S with one copy of (RS)₄; though the C-terminus of the RBD is also, like the others, enriched in serine and arginine.

RBSR2 (labelled as "WT" in Figs 10 and S12 and S13, and S5 and S6 Tables). Both RNAi lines had increased expression of variant surface glycoprotein genes that are normally silent, which might be a sign of stress. For both poly(A)+ (Fig 10C) and rRNA-depleted RNA (Fig 10D), the RBSR2 line showed preferential accumulation of shorter RNAs and loss of longer ones, but hardly any differences exceeded two-fold. The amount of mRNA, as judged by hybridization

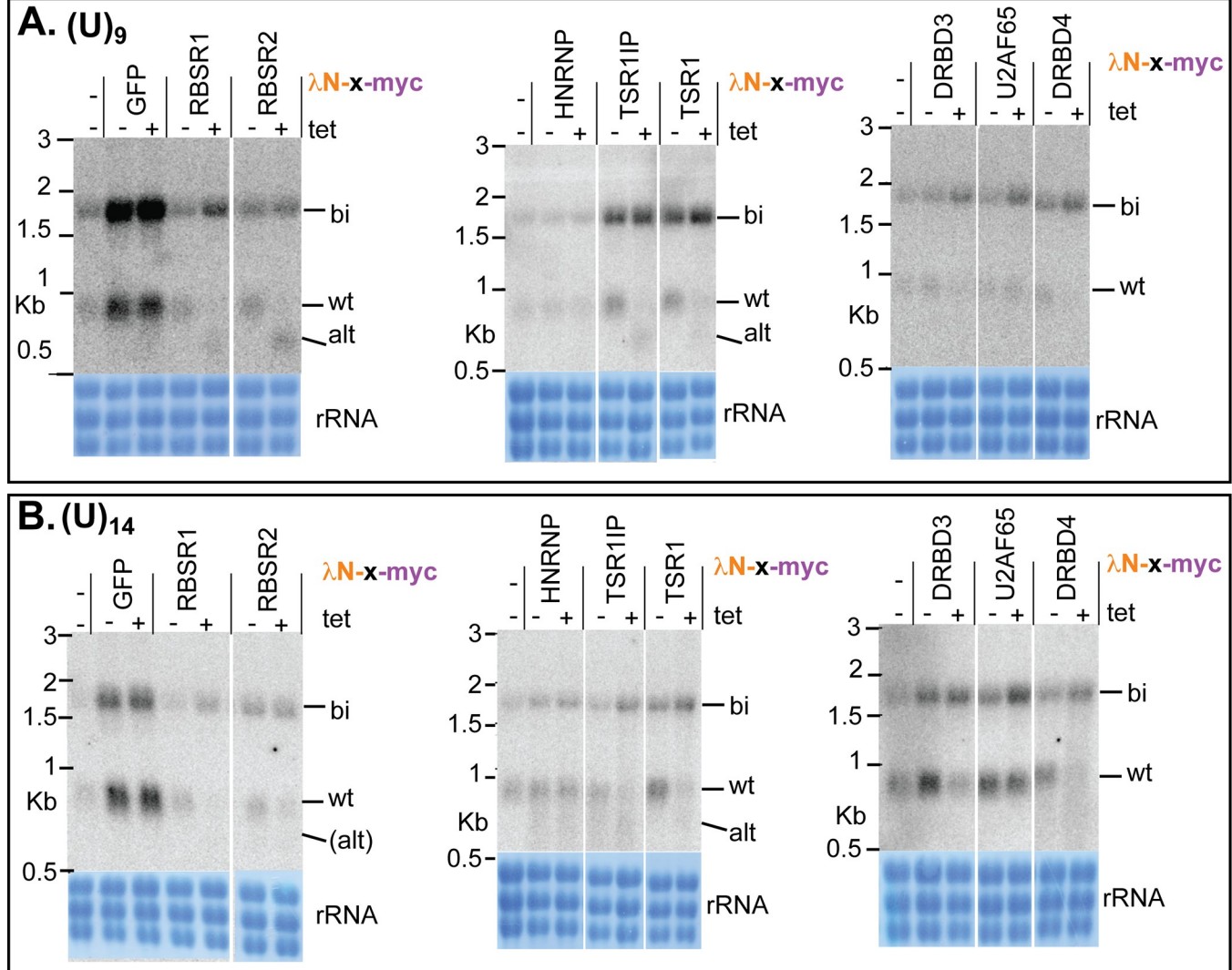

**Fig 8. Effects of tethering different proteins to boxB-(U)₉ and boxB-(U)₁₄ reporters.** Northern blots are shown for the *BSD* mRNAs. Additional clones and replicates are shown in S7–S9 Figs.

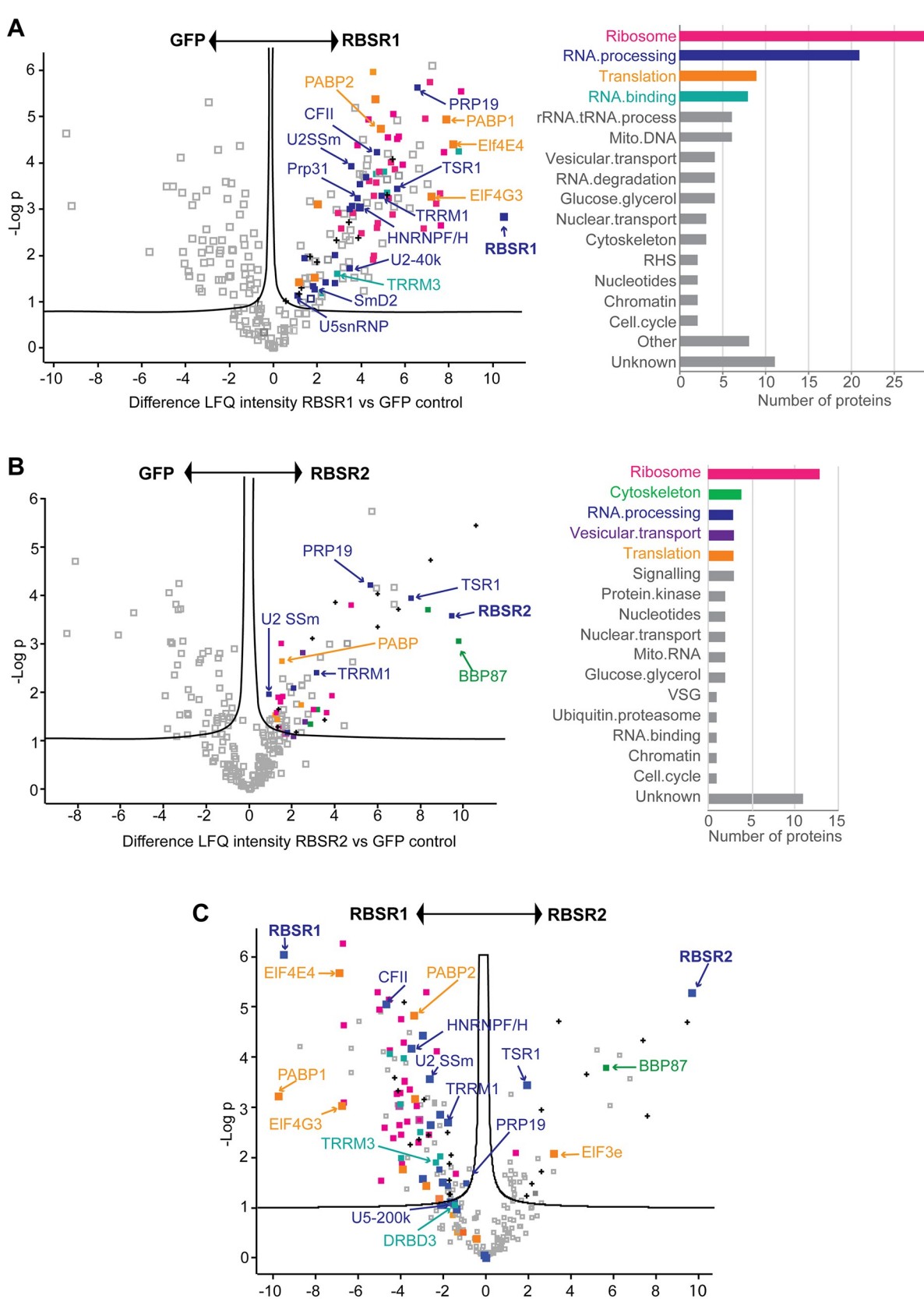

**Fig 9. Proteins associated with RBSR1 and RBSR2.** RBSR1-TAP and RBSR2-TAP were purified on IgG columns and released with TEV protease. Triplicate preparations were analyzed by mass spectrometry, using GFP-TAP as the control. Results were generated by PERSEUS. Proteins of interest are indicated. Functional categories were assigned manually from the annotation (see S3 Table). A. RBSR1 compared with GFP. Numbers of enriched proteins in particular categories are shown on the right. B. RBSR2 compared with GFP. Numbers of enriched proteins in particular categories are shown on the right. C. RBSR2 compared with RBSR1.

of a Northern blot with a spliced leader probe, was also twice that in the starting cell line. Since this effect is unrelated to the RBSR2 level and also to the cell density (S8 Fig), we cannot relate it to RBSR2 function. For RBSR1, we also noted small, mostly less than two-fold differences compared to the same "WT" cell line. For poly(A)+ mRNA, there was preferential loss of longer RNA (Fig 10E) but the correlation was weaker than for RBSR2, and absent for rRNA-depleted RNA (Fig 10F). Detailed examination of the read alignments for several genes that showed such discrepancies revealed no changes in polyadenylation or splice sites.

## Discussion

We here describe a vector that contains two selectable markers and can be used for detailed investigation of the effects of intergenic sequences on trypanosome gene expression (Fig 1). Since the precursor mRNA is rather inefficiently processed, both increases and decreases in processing can be measured or screened for. We here studied the effects of changing the PPT, which was the most obvious likely determinant. In the future it would be interesting to test a range of different intergenic regions for their effects on splicing and polyadenylation. In addition, our construct facilitates measurement of the effects of different 5'-UTRs on both processing and translation, without changing the upstream splicing signals.

The results in this paper confirm that a wide range of different PPTs can be used to specify *trans*-splicing acceptor sites in trypanosomes. When we screened 11mers in the PPT position (Fig 4C and S2 Table), nine pyrimidines were as effective in processing as ten or eleven pyrimidines. The preference for U over C was somewhat less pronounced than was seen in comparisons of sites used *in vivo* [23]. Including two boxB stem-loops just upstream of the PPT seemed to slightly inhibit processing directed by longer PPTs, but promote it with shorter ones. We have not investigated the reason for this but it perhaps suggests that the presence of a secondary structure can influence splice site choice. A study in yeast revealed that intron secondary structures can indeed influence splicing kinetics [71]. It is also possible that the two boxBs affect DNA structure and slow down RNA polymerase II elongation, enhancing splicing. In trypanosomes, RNA polymerase II tends to accumulate in regions preceding splice acceptor sites, perhaps indicating an elongation delay [72].

The vector that includes the boxB sequences was used to study the effects of proteins that are tethered just upstream of the PPT. The results of the experiments suggested that tethering was a useful, though not infallible, approach to investigate the functions of putative splicing factors. This is illustrated by our individual tests of processing factors. As expected, tethering of SF1 stimulated splicing at the "wild-type" site (Fig 5D). We had expected that tethering of CPSF3 would move both polyadenylation and splicing to downstream positions, but instead, it stimulated "correct" processing (Fig 5C). Presumably, tethered CPSF3 acted by recruiting splicing factors as well as the rest of the polyadenylation complex, but it is not clear why the "wild-type" acceptor was then used. Depletion of trypanosome U2AF65 inhibits splicing, and it interacts with SF1 but not U2AF35 [73]. Surprisingly, however, tethering of U2AF65 to the $(U)_3$ reporter weakly promoted alternative splicing, perhaps also increasing the amount of bicistronic RNA (Fig 5E). With the $(U)_9$ and $(U)_{14}$ reporters, no alternative splicing was seen but the bicistron was again more abundant (S7 Fig). Perhaps in this case, the tethered protein was impaired in splicing function.

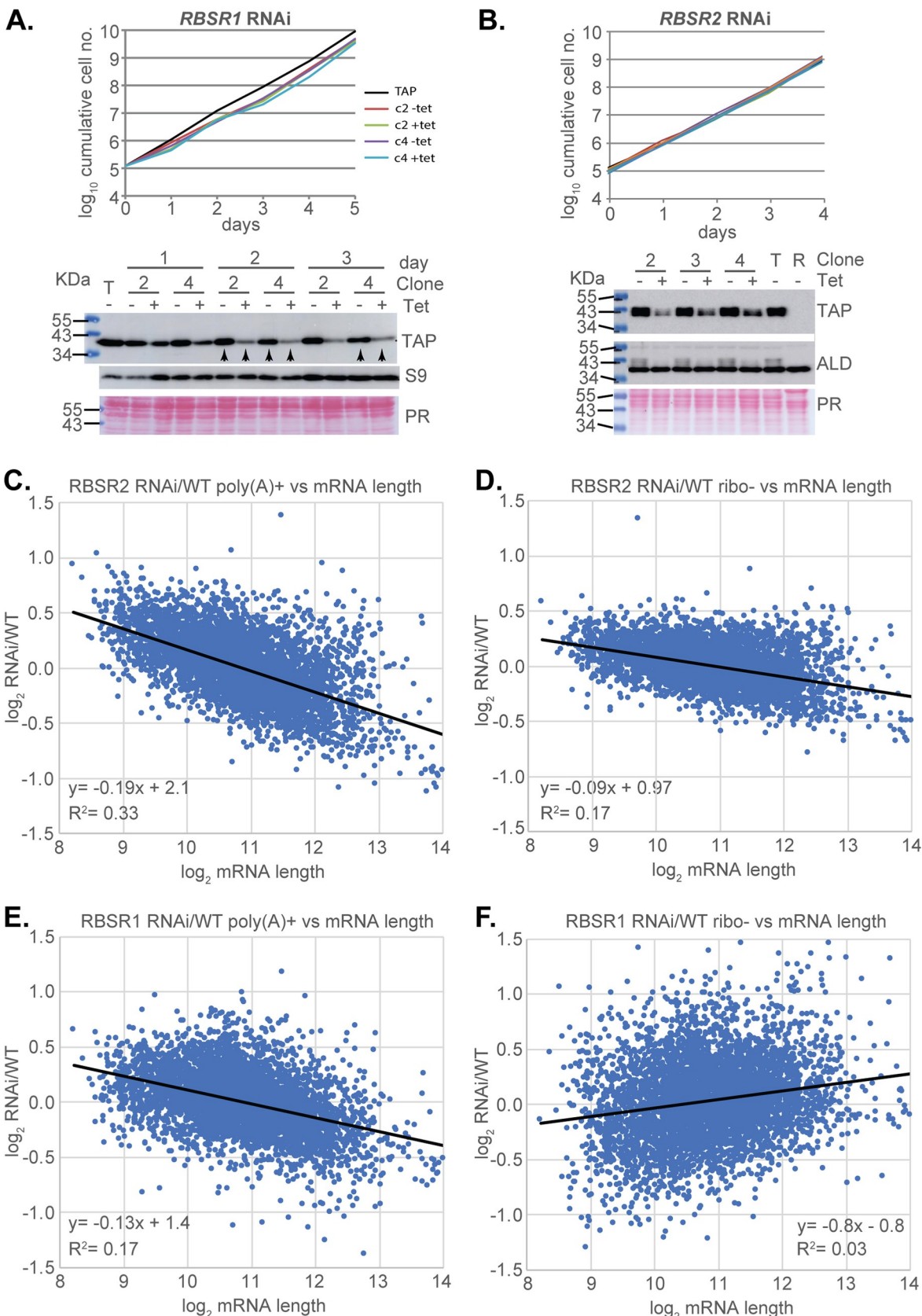

**Fig 10. Transcriptomes of trypanosomes containing integrated plasmids for RBSR1 and RBSR2 RNAi.** A. Growth of two clones (c2 and c4) with and without *RBSR1* RNAi induction. Cells with the tagged protein (TAP) are shown for comparison, and the Western blot is shown below. Further (uncropped) results are shown in S11 Fig. The arrows indicate the samples taken for RNASeq. B. Growth of two clones with and without *RBSR2* RNAi induction. Cells with the tagged protein (TAP) are shown for comparison, and the Western blot is shown below. The lines are not labelled because they are indistinguishable. Further (uncropped) results are shown in S12 Fig. C. Poly(A) + transcriptomes from the RBSR2 RNAi line, with and without tetracycline, compared with the "wild-type" (cells with tagged RBSR2). The ratio of RBSR2 line/WT is shown on the y-axis and the mRNA length on the x-axis. The formula for the regression line and the Pearson correlation coefficient are also shown. For more details, see S12 and S13 Figs, as well as S6 Table. D. rRNA-depleted transcriptomes from the RBSR2 RNAi line, with and without tetracycline, compared with the "wild-type" (cells with tagged RBSR2). The ratio of RBSR2 line/ WT is shown on the y-axis and the mRNA length on the x-axis. E. Poly(A)+ transcriptomes from the RBSR1 RNAi line, with and without tetracycline, compared with the "wild-type" (again, cells with tagged RBSR2, since the tagging had no effect on growth). The ratio of RBSR1 line/WT is shown on the y-axis and the mRNA length on the x-axis. F. rRNA-depleted transcriptomes from the RBSR1 RNAi line, with and without tetracycline, compared with the "wild-type" (cells with tagged RBSR2). The ratio of RBSR1 line/WT is shown on the y-axis and the mRNA length on the x-axis.

DRBD3 and DRBD4 [30,31,74] are the closest homologues of mammalian polypyrimidine-tract-binding proteins (PTB)), which have four RNA-recognition motifs (RRMs) [28]. DRBD3 has two RRMs while DRBD4 has four, although one is rather weak [30]. DRBD4 is predominantly found in the nucleus [30], whereas DRBD3 is concentrated in the nucleus but is also in the cytoplasm [30,31]. RNAi against either DRBD3 or DRBD4 caused both increases and decreases in transcript levels. Published results indicated that binding of DRBD3 to the 3'-UTR of an mRNA can, in at least some cases, increase mRNA stability [30,31,75], However, DRBD3 was also previously shown to pull down U1-70K and U1C, consistent with a role in splicing [75]. A conserved binding motif, UUCCCCUCU, was identified for DRBD3 [32,36]. In our PPT screen, this sequence was present in a single 11mer, but the read count numbers were too low to assess enrichment after selection so we do not know whether or not it is a good splicing signal.

Cumulative evidence suggests that mammalian PTB represses intron inclusion by binding either within the exon, or at the 3' splice site [28]. Suggested mechanisms of action include looping out the branch-point adenosine, or looping out entire exons through interaction with upstream and downstream PPTs [76]. Our results suggest that DRBD3 and DRBD4 may indeed have PTB-like functions: they both repressed *PAC trans* splicing when tethered, even when the reporter included a strong PPT (Figs 6 and S7). We do not know whether looping is required for DRBD3 and DRBD4 to inhibit processing. Their binding at PPTs that are normally used to direct splicing would repress mRNA expression, while binding within 3'-UTRs might, at least in some cases, act to maintain correct mRNA processing patterns, as recently observed for another RNA-binding protein, DRBD18 [77].

In Opisthokonts, SR-domain splicing regulators are known to define exon junctions during *cis*-splicing by binding at the beginning of the exon [29]. Although the kinetoplastid SR-domain proteins TSR1 and RBSR1 had previously been investigated [35,69,78], their functions in splicing were unknown. The effects of tethering trypanosome SR-domain proteins upstream of the PPT gave results that were entirely consistent with predictions from Opisthokonts. Tethering TSR1 and its interaction partner TSR1IP shifted splicing to the alternative upstream location even when a strong ((U)$_{14}$) PPT was present (Figs 7 and S8). Tethered RBSR1 and RBSR2 also shifted splicing upstream when the (U)$_3$ or (U)$_9$ reporters were used (Figs 7 and 8). With the (U)$_{14}$ reporter, use of the "wild-type" acceptor site was inhibited but the alternative products were not detected (Fig 8).

RBSR1 and RBSR2 had not previously been specifically investigated in *T. brucei*. Since the tethering result suggested that they might influence splicing, we followed up the prediction in further experiments. Depletion of either protein had little effect on cell proliferation, and transcriptome differences relative to a cell line without RNAi were unrelated to the level of the

target protein. A clear discrepancy between poly(A)+ and rRNA-depleted RNA in the cell line with *RBSR1* RNAi was intriguing but difficult to interpret. If it was caused by the 50% loss of RBSR1 in the uninduced line, why was it not more severe after RNAi induction? One possibility was that selection of the RNAi lines (two or three independent clones) had by itself caused transcriptome changes. Meanwhile, the proteins associated with RBSR1 and RBSR2 were much more instructive. RBSR1 preferentially copurified proteins associated with splicing, most particularly five components of the PRP19 complex, which is thought to associate with the spliceosome when the U4 snRNP dissociates from U5 and U6. This, and the presence of two U2 components, is consistent with RBSR1 affecting splice acceptor site choice. (Oddly, however, splicing factors were not detected in a previous non-quantitative study of proteins associated with RBSR1 in *Trypanosoma cruzi* [69].) Although RBSR1 is detected exclusively in the nucleus [68], RBSR1 also specifically pulled down one of the cap-binding translation initiation factor complexes, EIF4E4 and EIF4G3. The specificity is intriguing because two other translation initiation factor complexes, EIF4E3/EIF4G4 and EIF4E6/EIF4G5, are just as abundant [70]. Results from *Leishmania* suggest that the EIF4E4/G3 complex is implicated in translation of mRNAs encoding ribosomal proteins [79]. This link, as well as the apparent association of RBSR1 with the ribosome, might merit further investigation.

RBSR2 was associated with only three splicing-related proteins: TSR1, TRRM1 and PRP19. It is thus possible that the effect of tethered RBSR2 on splice site choice was due to its association with TSR1. Intriguingly, however, we also detected specific associations with proteins of the basal body. Inspection of available images on the tryptag.org web site [68] shows that in addition to predominantly nuclear localization, GFP-tagged RBSR2 gives diffuse cytoplasmic fluorescence and sometimes also one or two spots, but the location of the latter is not consistent with basal body association. RBSR2 also pulled down the three major vault proteins, which are all cytoplasmic when GFP-tagged [68]. The function of vault proteins in trypanosomes is unknown.

In conclusion, our results show that our vector system is useful to examine the sequence determinants for efficient mRNA processing. We also demonstrated that tethering of potential splicing regulators upstream of a *trans* splice acceptor site can give insights into factor function. Our vectors could be used to screen for novel splicing factors with either stimulating or inhibitory functions although—as for all screens—results would have to be confirmed individually and predictions would have to be tested by other methods. We further expect that similar systems could be used for studies of both *cis* and *trans* splicing in other organisms.

## Supporting information

**S1 Fig. Raw data for Figs 5 and 6: Western blot, full Northern blots with replicates for two clones and puromycin susceptibility curves labelled according to clone, for tethering of GFP and HNRPH/F.**
(TIF)

**S2 Fig. Raw data for Fig 5: Western blot, full Northern blots with replicates for two clones. and puromycin susceptibility curves labelled according to clone, for tethering of CPSF3 and SF1.**
(TIF)

**S3 Fig. Raw data for Fig 5: Western blot, full Northern blots with replicates for two clones. and puromycin susceptibility curves labelled according to clone, for tethering of CPSF65.**
(TIF)

**S4 Fig. Raw data for Fig 6: Western blot, full Northern blots with replicates for two clones. and puromycin susceptibility curves labelled according to clone, for tethering of DRBD3 and DRBD4.**
(TIF)

**S5 Fig. Western blot, full Northern blots with replicates for two clones. and puromycin susceptibility curves labelled according to clone, for tethering of TSR1 and its interaction partner TSRIP.**
(TIF)

**S6 Fig. Western blot, full Northern blots with replicates for two clones, and puromycin susceptibility curves labelled according to clone, for tethering of RBSR1 and RBSR2.**
(TIF)

**S7 Fig. Western blot, full Northern blots for *BSD* and *PAC*, with replicates for two clones, for tethering of DRBD3, U2AF65 and DRBD4 to the $(U)_9$ and $(U)_{14}$ boxB reporters.**
(TIF)

**S8 Fig. Western blot, full Northern blots for *BSD* and *PAC*, with replicates for two clones, for tethering of HNRNPF/H, TSR1 and TSRIP to the $(U)_9$ and $(U)_{14}$ boxB reporters.**
(TIF)

**S9 Fig. Western blot, full Northern blots for *BSD* and *PAC*, with replicates for two clones, for tethering of RBSR1 and RBSR2 to the $(U)_9$ and $(U)_{14}$ boxB reporters.**
(TIF)

**S10 Fig. Western blots showing levels of RBSR1-TAP with and without RNAi induction.** "P" is the precursor cell line with the TAP tag but no RNAi plasmid. The numbers are quantitation and the arrows indicate samples used for RNASeq.
(TIF)

**S11 Fig. Western blots showing levels of RBSR2-TAP with and without RNAi induction are in A and B.** "P" is the precursor cell line with the TAP tag but no RNAi plasmid. The numbers are quantitation and the arrows indicate samples used for RNASeq. S9 is ribosomal protein S9. Panel C shows a Northern blot of the RNA used for seequencing, hybridised with a spliced leader probe. "P? is two samples of the input (tagged) line that serves as the "wild-type" control in the RNASeq analysis. "wt" is RNA from another experiment; the low amount of mRNA in these might be caused by high cell density but this is uncertain. Two exposures are shown and relative quantitation of the shorter exposure (normalised to rRNA) is shown.
(TIF)

**S12 Fig. Principal component analysis for RNASeq of the RBSR1 and RBSR2 RNAi lines.** "WT" here refers to the line expressing RBSR2-TAP, without any RNAi plasmid. "m" means minus tetracycline, "p" means plus tetracycline, for the three replicates illustrated in S10 and S11 Figs. Cell densities (multiplied by $10^{-5}$) and the percent of the RBSR protein for each sample (as shown in S10 and S11 Figs) are also shown. "pA" is poly(A)+ RNA, ribominus is rRNA-depleted RNA.
(PDF)

**S13 Fig. Scatter plots comparing the RNASeq datasets.** In all cases the results for +tet and -tet were pooled. All results are log2-transformed. A. RBSR2, RNAi cell line/WT, poly(A)+, on x-axis, ribo-minus on y-axis. B. RBSR2, RNAi cell line/WT, poly(A)+, coding sequence (CDS) on x-axis, 3'-UTR on y-axis. C. RBSR1, RNAi cell line/WT, poly(A)+, on x-axis, ribo-minus on

y-axis. D. RBSR1, RNAi cell line/WT, poly(A)+, coding sequence (CDS) on x-axis, 3'-UTR on y-axis. E. RBSR1, RNAi cell line/WT, poly(A)+ result divided by ribo-minus result R on y-axis, mRNA length on x-axis.
(PDF)

**S1 Table. Plasmids and oligonucleotides.**
(XLSX)

**S2 Table. PPT screen.** For details see first sheet.
(XLSX)

**S3 Table. Mass spectrometry data.** For details see first sheet.
(XLSX)

**S4 Table. Abundances of proteins implicated in splicing, downloaded from [70].**
(XLSX)

**S5 Table. Transcriptome results: raw reads.**
(XLSX)

**S6 Table. Transcriptome results: DeSeq2 analysis.**
(XLSX)

**S1 Text. Intergenic sequences used for Fig 3.**
(DOCX)

**S1 File. pHD3180 sequence, ApE format.**
(APE)

**S2 File. pHD3186 sequence, ApE format.**
(APE)

**S3 File. pHD3190 sequence, ApE format.**
(APE)

**S4 File. pHD3259 sequence, ApE format.**
(APE)

## Acknowledgments

We thank Claudia Helbig and Ute Leibfried for technical assistance, and Claudia Helbig especially for several of the Northern blots. We also thank David Ibberson of the BioQuant sequencing facility (University of Heidelberg) for DNA library construction and sequencing. We thank Shula Michaeli for many useful discussions and Andreas Kulozik and Susanne Kramer for suggestions during advisory meetings. We are indebted to Nina Papavasiliou (DKFZ, University of Heidelberg) and Luise Krauth-Siegel (BZH, University of Heidelberg) for allowing us to share their laboratories including equipment and reagents after the flood in the ZMBH.

## Author Contributions

**Conceptualization:** Albina Waithaka, Christine Clayton.

**Formal analysis:** Albina Waithaka, Olena Maiakovska, Christine Clayton.

**Funding acquisition:** Dirk Grimm, Christine Clayton.

**Investigation:** Albina Waithaka.

**Methodology:** Albina Waithaka, Olena Maiakovska, Larissa Melo do Nascimento, Christine Clayton.

**Project administration:** Christine Clayton.

**Software:** Olena Maiakovska.

**Supervision:** Dirk Grimm, Larissa Melo do Nascimento, Christine Clayton.

**Visualization:** Albina Waithaka, Christine Clayton.

**Writing – original draft:** Albina Waithaka, Christine Clayton.

**Writing – review & editing:** Albina Waithaka, Olena Maiakovska, Dirk Grimm, Christine Clayton.

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
