## [Decision Letter · Decision Letter 0]

5 Sep 2022

Dear Prof. Clayton,

Thank you very much for submitting your manuscript "Sequences and proteins that influence mRNA processing in Trypanosoma brucei: evolutionary conservation of SR-domain and PTB protein functions" for consideration at PLOS Neglected Tropical Diseases. As with all papers reviewed by the journal, your manuscript was reviewed by members of the editorial board and by several independent reviewers. The reviewers appreciated the attention to an important topic. Based on the reviews, we are likely to accept this manuscript for publication, providing that you modify the manuscript according to the review recommendations. 

Sincerely,

Paul O. Mireji, PhD

Academic Editor

Charles Jaffe

Section Editor

Reviewer's Responses to Questions

**Key Review Criteria Required for Acceptance?**

**Methods**

-Are the objectives of the study clearly articulated with a clear testable hypothesis stated?

-Is the study design appropriate to address the stated objectives?

-Is the population clearly described and appropriate for the hypothesis being tested?

-Is the sample size sufficient to ensure adequate power to address the hypothesis being tested?

-Were correct statistical analysis used to support conclusions?

-Are there concerns about ethical or regulatory requirements being met?

Reviewer #1: The objectives of the study clearly articulated in lines 109 -110 and the study design appropriately addresses the stated objectives. However, it adds clarity to provide a citation or document how the radiolabelled probes were generated (line 146).

Reviewer #2: (No Response)

Reviewer #3: The authors report on investigations on factors which influence the choice of site for trans-splicing of a capped leader sequence at the 5’end of mRNA in Trypanosoma brucei. For the investigations, the authors used a transgenic clone of T. brucei maintained by growth in culture. 

The paper is in part a methods one, in which the authors describe in detail (their) approaches to finding polypyrimmidine tracts (PPTs) active in the trans-splicing in trypanosomes. Since the model used (transgenic L427) is a trypanosome clone which has been maintained in culture for many years and which, therefore, probably can no longer grow in the mammalian host, the authors should be cautious in extrapolating observations made here to the trypanosomes growing in the mammalian host. Alibu et al. does not appear in the list of references (line 116).

**Results**

-Does the analysis presented match the analysis plan?

-Are the results clearly and completely presented?

-Are the figures (Tables, Images) of sufficient quality for clarity?

Reviewer #1: The analyses presented are ideal for the hypotheses tested and the results are adequately presented. All the figures (Tables, Images) are of sufficient quality and clarity. RBSR1 and RBSR2 (Line 409) should be TbRBSR1 and TbRBSR2 respectively especially since the current study confirmed their existance in T. brucei. The entire reporter construction section spanning lines 182 to 198 is more of a method and less of a result and should be placed correctly within the methods section especially considering that the PPT library preparation section depends on the constructed pHD3180 plasmid.

Reviewer #2: (No Response)

Reviewer #3: The results are clearly presented, and the figures are of reasonable quality. However, the authors should explain, in the appropriate section of the manuscript, the differences in average size of the RNA in Fig. 3, lanes 1and 2, ACT, BSD. They seem to be identical in size under PAC (line 284). Furthermore, there should be an explanation of the bases upon which the predictions were made regarding the faster than average splicing, and slower processing of some transcripts referred to here (lines 294-295). What is the statistic supporting the significance of level of homology? Either cite a reference or provide an explanation

**Conclusions**

-Are the conclusions supported by the data presented?

-Are the limitations of analysis clearly described?

-Do the authors discuss how these data can be helpful to advance our understanding of the topic under study?

-Is public health relevance addressed?

Reviewer #1: All the conclusions as drawn are sufficiently supported by the data presented (Figures 1-10 and S1-S13; Tables S1-SS5.

Reviewer #2: (No Response)

Reviewer #3: The discussion is too long and rambles on without clearly bringing out the key contribution of the work. It would be most helpful to state concisely and discus the key features of this potentially useful vector, and its general applicability in studying trans-splicing in trypanosomes, etc. In particular, there is only scanty reference to figures which may be relevant to the point(s) being discussed. This leaves it to the reader to guess which figure the authors have in mind when making a particular point in the Discussion.

**Editorial and Data Presentation Modifications?**

Reviewer #1: The various Trypanosoma brucei homologues of splicing factors or regulators (CPSF3, SF1, DRBD3, DRBD4, RBSR1, RBSR2,TSR1,HNRNPF/H, U2AF65) should be clearly named in Lines 109 -110 such that a reader is not left guessing.

Reviewer #2: (No Response)

Reviewer #3: Accept wit minor revision.

**Summary and General Comments**

Reviewer #1: The section that begins"Two Life cycles stages of T. brucei grow well in vitro: the mammalian "bloodstream" form, and the "procyclic" form which grows in the midgut of tsetse flies is misplaced and should be positioned around or just before line 70.

Delete "that" appearing at the beginning of line 261.

Reviewer #2: Kinetoplastida transcribe mRNAs polycistronically and process the polycistrons by trans splicing, with the splicing process being coupled to the polyadenylation of the upstream mRNA. A polypyrimidine tract PPT is important, and the length of the sequence between the splice site and the upstream pA site seems conserved. However, sequences required for the splicing appear not very conserved and alternative PPT can be used, and frequently are (resulting also in alternative adenylations).

The aim of the authors was to establish a screening system that allows to screen for both cis-acting factors (e.g. the sequence of the PPT) and trans-acting factors that have positive or negative effects on splicing efficiency in Trypanosoma brucei. The authors have constructed a reporter system out of two different antibiotica resistance genes, connected by a “linker” that is a hybrid of the intergenic regions of two different genes. They then included a B-box, to allow tethering of proteins with a suspected function in splicing. This dicistron has relatively poor splicing efficiency, with both the dicistron and the monocistrons being readily detectable on a northern blot. This allows to screen for factors that either increase or decrease splicing efficiency. The authors have carefully established the system, using positive and negative controls. They tested cis-elements in a large screen and tethered a few proteins. All experiments are well performed and some meaningful data are produced. 

The system has its weaknesses, in particular for larger screens, and the authors are aware of it. Tethering data are always a bit tricky as the orientation matters. The output (antibiotica resistance) is not only determined by splicing efficiency, but can also be affected by differences in export and translation efficiency and it is not well known how such dicistronic transcripts behave in trypanosomes in this regard. This means, that the system still requires northern blots for data interpretation, which are not high throughput. For these reasons, this reviewer is not convinced that the system is very suitable for further screens. However, the data are still highly interesting, as they contain a few conclusions about splice sites in trypanosomes that are new. They support the current model that trypanosomes often “find a way” to splice their mRNAs even with sub-optimal splice sites. And they provide a tool to produce dicistronic mRNAs, perhaps for further studies of the subcellular localization and mRNA export control in trypanosomes.

I do have a few comments:

Figure 1 C-E

It would be easier for the reader, if the authors could include the data from E to C and D, rather than showing a wrong clone (its already rather complex). The difficulties to get the clone can be discussed in the text (as they are meaningful) and perhaps the data can also be shown in sup. Material.

Figure 1, C and D

I would expect that the splicing efficiency with the same PTT is equal, independent on whether the Northern blot is probed for BSD or PAC. But it looks like PAC monomer per BSD-PAC dimer is a lot more than BSD monomer per BSD-PAC dimer. Can the authors provide an explanation? Is the BSD mRNA much less stable than the PAC mRNA? And if so, how does it impact the screen?

The authors show that the dicistronic RNA gives poor blasticidin resistance, possibly because the PAC sequence in the ‘UTR’ interferes with either translation or nuclear export. Still, they use blasticidin selection in their cis-element PPT screen to obtain the clones, that are then, in a second step, challenged with puroymcin selection to look for splicing supportive or suppressive sequences. How can the authors be sure that splicing-suppressive sequences are not lost in the blastidicin selection step already?

RBSR1 / RBSR2 dataset: It doesn’t really fit into the story... Consider to publish as individual manuscript? Or, if not possible, restructure a little bit, last part of the manuscript application to the unknown, 1) interaction data and 2) test for splicing efficiency. 

Susanne Kramer

Reviewer #3: The authors report on investigations on factors which influence the choice of site for trans-splicing of a capped leader sequence at the 5’end of mRNA in Trypanosoma brucei. For the investigations, the authors used a transgenic clone of T. brucei maintained by growth in culture. The manuscript is well-written and is sufficiently detailed, only requiring a few improvements, as suggested.

PLOS authors have the option to publish the peer review history of their article (what does this mean?). If published, this will include your full peer review and any attached files.

Reviewer #1: No

Reviewer #2: Yes: Susanne Kramer

Reviewer #3: Yes: Phelix Majiwa

Figure Files:

Data Requirements:

Reproducibility:

References

---

## [Decision Letter · Decision Letter 1]

7 Oct 2022

Dear Prof. Clayton,

We are pleased to inform you that your manuscript 'Sequences and proteins that influence mRNA processing in *Trypanosoma brucei*: evolutionary conservation of SR-domain and PTB protein functions' has been provisionally accepted for publication in PLOS Neglected Tropical Diseases.

Best regards,

Paul O. Mireji, PhD

Academic Editor

Charles Jaffe

Section Editor

Reviewer's Responses to Questions

**Key Review Criteria Required for Acceptance?**

**Methods**

-Are the objectives of the study clearly articulated with a clear testable hypothesis stated?

-Is the study design appropriate to address the stated objectives?

-Is the population clearly described and appropriate for the hypothesis being tested?

-Is the sample size sufficient to ensure adequate power to address the hypothesis being tested?

-Were correct statistical analysis used to support conclusions?

-Are there concerns about ethical or regulatory requirements being met?

Reviewer #1: Corrections adequately done on resubmitted version.

Reviewer #2: (No Response)

Reviewer #3: The Methods section is clearer in the revised version and meets all the criteria stipulated above.

**Results**

-Does the analysis presented match the analysis plan?

-Are the results clearly and completely presented?

-Are the figures (Tables, Images) of sufficient quality for clarity?

Reviewer #1: Now in tandem with my recommendations.

Reviewer #2: (No Response)

Reviewer #3: The Results section in the Revised Version is more detailed, and contains the relevant additional information to satisfy this reviewer's questions.

**Conclusions**

-Are the conclusions supported by the data presented?

-Are the limitations of analysis clearly described?

-Do the authors discuss how these data can be helpful to advance our understanding of the topic under study?

-Is public health relevance addressed?

Reviewer #1: Done.

Reviewer #2: (No Response)

Reviewer #3: The Conclusions reached by the authors, as presented in the Discussion section, are valid. The Discussion section is now more focused and clearer compared to what it was in the earlier/first version.

**Editorial and Data Presentation Modifications?**

Reviewer #1: Acceptable.

Reviewer #2: (No Response)

Reviewer #3: I recommend to Accept the current revised version of the Manuscript.

**Summary and General Comments**

Reviewer #1: Adequate and in sync with the results presented.

Reviewer #2: I m happy with the manuscript now.

Reviewer #3: This Reviewer has nothing useful to add.

PLOS authors have the option to publish the peer review history of their article (what does this mean?). If published, this will include your full peer review and any attached files.

Reviewer #1: **Yes: **Benson Nyambega

Reviewer #2: No

Reviewer #3: **Yes: **Phelix A.O. Majiwa

---

## [Editor Report · Acceptance letter]

21 Oct 2022

Dear Prof. Clayton,

We are delighted to inform you that your manuscript, "Sequences and proteins that influence mRNA processing in *Trypanosoma brucei*: evolutionary conservation of SR-domain and PTB protein functions," has been formally accepted for publication in PLOS Neglected Tropical Diseases.

Best regards,

Shaden Kamhawi

co-Editor-in-Chief

Paul Brindley

co-Editor-in-Chief
